# EXPLORING NON-CONVEX DISCRETE ENERGY LANDSCAPES: A LANGEVIN-LIKE SAMPLER WITH REPLICA EXCHANGE

## ABSTRACT

Gradient-based Discrete Samplers (GDSs) are effective for sampling discrete energy landscapes. However, they often stagnate in complex, non-convex settings. To improve exploration, we introduce the Discrete Replica EXchangE Langevin (DREXEL) sampler and its variant with Adjusted Metropolis (DREAM). These samplers use two GDSs (Zhang et al., 2022b) at different temperatures and step sizes: one focuses on local exploitation, while the other explores broader energy landscapes. When energy differences are significant, sample swaps occur, which are determined by a mechanism tailored for discrete sampling to ensure detailed balance. Theoretically, we prove both DREXEL and DREAM converge asymptotically to the target energy and exhibit faster mixing than a single GDS. Experiments further confirm their efficiency in exploring non-convex discrete energy landscapes.

## 1 INTRODUCTION

Sampling from high-dimensional discrete distributions has been an important task for decades across applications in texts (Mikolov et al., 2013; Devlin et al., 2019), images (Krizhevsky et al., 2012; Ronneberger et al., 2015), signal processing (Mallat, 1989; Donoho, 2006), genome sequences (Metzker, 2010; Macosko et al., 2015), etc. However, the exponential growth in the number of configurations makes sampling from $\pi(\theta) \propto \exp[U(\theta)]$ computationally prohibitive. The computational burden comes from evaluating the exact probabilities and normalizing constants, which makes exact sampling impossible in practice. Algorithms such as rejection sampling (Neumann, 1951), Swendsen-Wang (Swendsen & Wang, 1987), and Hamze-Freitas (Hamze & de Freitas, 2004) leverage special structures within the problem to make global updates. In more general settings, these methods may suffer from slow exploration, local dependencies, and poor convergence.

To make high-dimensional discrete sampling more efficient, Locally Balanced Proposals (LBPs) (Zanella, 2020; Sun et al., 2021) improved acceptance rates by adjusting proposal distributions based on the likelihood ratio. Early LBPs updated one coordinate at a time (Zanella, 2020; Grathwohl et al., 2021), and Grathwohl et al. (2021) developed gradient-based discrete sampler (GDS) to update coordinately. Later, Zhang et al. (2022b) further extended GDSs by updating all coordinates simultaneously, which enhances efficiency and scalability for large-scale, high-dimensional computations on GPUs and TPUs.

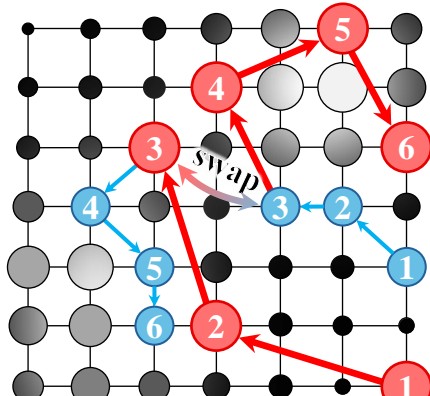

Figure 1: DREXEL & DREAM sample trajectory in discrete domains. Blue denotes a low-temperature sampler, and red high-temperature sampler. They exchange samples following a swap mechanism.

Despite improvements in LBPs, how to balance the trade-off between "global exploration" and "local exploitation" remains a challenge. High-dimensional discrete distributions are highly multi-modal, with deep and narrow wells caused by intrinsic discontinuities. Gradient-based LBPs, although effective, tend to get trapped in local modes due to their reliance on local gradients and small noise, which is insufficient for escaping these traps.

To bridge this gap, we propose two samplers: Discrete Replica EXchangE Langevin (DREXEL) and Discrete Replica Exchange with Adjusted Metropolis (DREAM). These samplers combine

GDS with the replica exchange Markov Chain Monte Carlo (reMCMC) (Chen et al., 2019) for efficient exploration of non-convex discrete spaces. As illustrated in Figure 1, the samplers employ two GDSs at different temperatures and step sizes: the low-temperature sampler focuses on local exploitation, while the high-temperature sampler escapes local traps for broader exploration. Sample swaps occur when energy differences are significant, governed by a mechanism tailored for discrete sampling to ensure detailed balance. The combination of replica exchange and GDS makes them particularly effective for sampling from complex discrete structures in modern applications. The primary contributions in this work are summarized as follows:

- A novel integration of GDS with replica exchange to improve non-convex exploration;
- A swap mechanism tailored for detailed balance and sample efficiency in discrete sampling;
- Theoretical analysis of improved mixing rates over naïve discrete Langevin-like samplers;
- Superior performance in synthetic tasks, Ising models, restricted Boltzmann machines, and energy-based deep learning models for navigating non-convex discrete energy landscapes.

## 2 RELATED WORK

**Gradient-based Discrete Sampling** becomes popular for complex discrete sampling tasks and its original idea comes from LBPs. The concept of LBPs, as introduced by Zanella (2020), utilized local information in the form of density ratios to improve the sample efficiency. Grathwohl et al. (2021) expanded LBPs by the use of first-order Taylor approximation, which further ensures computational feasibility. To improve sampling in high-dimensional discrete spaces, LBPs were extended to cover larger neighborhoods by performing a sequence of small moves (Sun et al., 2021). Zhang et al. (2022b) further developed GDSs, which adapt the continuous Langevin MCMC to discrete spaces and allow parallel updates of all coordinates based on gradient information. Subsequently, GDSs were improved through the introduction of an adaptive mechanism by which the step size can be automatically adjusted (Sun et al., 2023a). Most recently, Pynadath et al. (2024) introduced an automatic cyclical scheduling approach in step sizes to better handle multi-modal distributions by alternating between exploration and exploitation phases. While these methods have shown promise, sampling from highly non-convex discrete distributions remains challenging, particularly when dealing with strongly correlated variables or energy-based deep learning models.

**Replica Exchange** MCMC is a powerful method that enhances exploration in complex, multi-modal distributions, and a variety of related algorithms build on this. For instance, unadjusted Langevin MCMC (Durmus & Moulines, 2017) leverages gradient information to guide proposals but lacks the exchange mechanism. Importance sampling (Wang & Landau, 2001) adjusts for the discrepancy between target and proposal distributions, which offers flexibility in sampling but without temperature-based exchanges. Simulated tempering (Lee et al., 2018) further refined the temperature-scaling strategy by dynamically adjusting the temperature of a single chain. Recently, Zhang et al. (2020) proposed a cyclical step-size scheduler to maintain a balance between exploration and exploitation. To enhance the exploration, reMCMC runs multiple chains at different temperatures and allows for chain swaps between them. Dong & Tong (2022) analyzed its mixing by quantifying the spectral gap, and Deng et al. (2020; 2022) validated its efficiency in large-scale deep learning tasks. Despite its success in continuous sampling and high-dimensional settings, to the best of our knowledge, its potential has not been studied in sampling from discrete distributions.

## 3 PRELIMINARIES

**The Target Distribution** $\pi := \Theta \rightarrow [0, 1]$ denotes the probabilistic model we are sampling from:

$$\pi(\boldsymbol{\theta}) = \frac{1}{Z} \exp\left[\frac{U(\boldsymbol{\theta})}{\tau}\right], \quad \forall \boldsymbol{\theta} \in \Theta, \ \Theta \subset \mathbb{R}^{\mathbf{d}}. \tag{1}$$

Here $\boldsymbol{\theta}$ is $\mathbf{d}$-dimensional variable, $\tau = 1.0$ denotes the respective temperature, $\Theta$ is a finite domain, $U$ represents the energy function, and $Z$ normalizes the distribution. Following the traditional settings in discrete sampling, we assume:

1. The sampling domain is coordinate-wisely factorized where $\Theta = \prod_{d=1}^{\mathbf{d}} \Theta_d$, and we primarily consider the binary cases $\Theta = \{0, 1\}^{\mathbf{d}}$ or categorical $\{0, 1, \ldots, N - 1\}^{\mathbf{d}}$;
2. The energy function is differentiable across $\mathbb{R}^{\mathbf{d}}$.

The primary goal is to design an efficient sampler to approximate $\pi$ within a finite sample size. The empirical distribution derived from these samples converges to $\pi$, with the approximation error bounded by a constant $\epsilon > 0$ under specific metrics.

**Replica Exchange** MCMC is a popular sampling method for non-convex exploration in continuous spaces. It updates according to the following dynamics:

$$\boldsymbol{\theta}_{i+1}^{(k)} = \boldsymbol{\theta}_i^{(k)} + \frac{\alpha_k}{2} \nabla U(\boldsymbol{\theta}_i^{(k)}) + \sqrt{\alpha_k \tau_k} \xi_k, \quad k = 1, 2; \quad i = 1, 2, \cdots, I. \tag{2}$$

Here $\alpha_1, \alpha_2 \in \mathbb{R}^+$ represent step sizes, $\tau_1, \tau_2 \in \mathbb{R}^+$ are temperatures, and $\xi_1, \xi_2$ are independent Gaussian noises drawn from $\mathcal{N}(\mathbf{0}, \mathbf{I}_{\mathbf{d} \times \mathbf{d}})$. The typical setup assumes $\alpha_1 < \alpha_2$ and $\tau_1 < \tau_2$, with the first chain in (2) labeled as the low-temperature chain and the second as the high-temperature chain. The gradient $\nabla U(\cdot)$ directs the algorithm toward high-probability regions. To further improve the mixing rate over Langevin MCMC, reMCMC facilitates interaction via a chain swap mechanism. Specifically, the probability to swap the $i$-th samples between $\boldsymbol{\theta}_i^{(1)}$ and $\boldsymbol{\theta}_i^{(2)}$ is determined by $\rho \min \left\{ 1, \ S(\boldsymbol{\theta}_i^{(1)}, \boldsymbol{\theta}_i^{(2)}) \right\}$. The swap intensity is regulated by $\rho > 0$, and the swap function $S := \Theta \times \Theta \to \mathbb{R}^+$ is given as follows:

$$S(\boldsymbol{\theta}^{(1)}, \boldsymbol{\theta}^{(2)}) = e^{\left(\frac{1}{\tau_2} - \frac{1}{\tau_1}\right)\left[U(\boldsymbol{\theta}^{(1)}) - U(\boldsymbol{\theta}^{(2)})\right]}. \tag{3}$$

Intuitively, the swap probability in reMCMC depends on the energy estimated at $\boldsymbol{\theta}^{(1)}$ and $\boldsymbol{\theta}^{(2)}$. When the low-temperature chain gets stuck in a local minimum and the high-temperature chain escapes to find modes with significantly lower energy, the chain will swap their samples with high probability. This enables the low-temperature chain to better characterize the newly discovered modes, while the high-temperature chain continues to search across the energy landscape. As mentioned in Chen et al. (2019), reMCMC behaves as a reversible Markov jump process due to its swap mechanism, which converges to a similar invariant distribution in (1) while the parameters can explore over $\mathbb{R}^{\mathbf{d}}$.

While reMCMC is a powerful tool for non-convex exploration, its update may fail to preserve the target distribution due to discretization errors introduced by the finite step size (Welling & Teh, 2011). According to Roberts & Tweedie (1996), selecting an inappropriate step size can lead to a transient Markov chain without a stationary distribution. To mitigate such bias, two main approaches are commonly used: decaying step sizes (Vollmer et al., 2016; Teh et al., 2016) and Metropolis-Hastings (MH) corrections (Dwivedi et al., 2019; Chewi et al., 2021). While implementing decaying step sizes is straightforward and does not require additional computational burden, the second approach is more favorable due to its specific advantages in discrete sampling, which will be elaborated on later.

**Metropolis-Hastings Correction** is considered to correct discretization errors and ensure convergence to the target distribution. Specifically, at each iteration, a new candidate $\omega \leftarrow \boldsymbol{\theta}_{i+1}$[1] is first generated with (2). To ensure that the resulting samples come from the target distribution, the MH step determines whether to accept or reject the candidate with $\mathcal{A} := \Theta \times \Theta \to [0, 1]$:

$$\mathcal{A}(\omega, \boldsymbol{\theta}_i) = \min \left\{ 1, \frac{\pi(\omega) q(\boldsymbol{\theta}_i \mid \omega)}{\pi(\boldsymbol{\theta}_i) q(\omega \mid \boldsymbol{\theta}_i)} \right\}, \tag{4}$$

where $q := \Theta \times \Theta \to [0, 1]$ is the transition probability mapping from the current sample $\boldsymbol{\theta}_i$ to the next sample $\boldsymbol{\theta}_{i+1}$. With probability $\mathcal{A}(\omega, \boldsymbol{\theta}_i)$, the candidate $\omega$ is accepted in the current step; otherwise, it retains the current $\boldsymbol{\theta}_i$. This adjustment preserves the correct stationary distribution. Furthermore, because Langevin MCMC allows each sample to access any point in $\mathbb{R}^{\mathbf{d}}$, it further ensures the Markov chain is both irreducible and ergodic (Diaconis & Freedman, 1997; Meyn & Tweedie, 2012).

**Discrete Langevin Sampler** (DLS) is a gradient-based approach for sampling from high-dimensional discrete distributions. Inspired by Langevin MCMC, DLS updates all coordinates in parallel from a single gradient computation to function effectively in discrete settings. Specifically, for a target distribution $\pi \propto \exp[U(\cdot)]$, DLS generates a new sample $\boldsymbol{\theta}_{i+1}$ inspired by the Taylor expansion:

$$q(\boldsymbol{\theta}_{i+1} \mid \boldsymbol{\theta}_i) = \frac{\exp\left(-\frac{1}{2\alpha} \left\| \boldsymbol{\theta}_{i+1} - \boldsymbol{\theta}_i - \frac{\alpha}{2\tau} \nabla U(\boldsymbol{\theta}_i) \right\|_2^2\right)}{Z_{\Theta}(\boldsymbol{\theta}_i)}, \quad \boldsymbol{\theta}_i, \boldsymbol{\theta}_{i+1} \in \Theta. \tag{5}$$

---

[1]For clarity and conciseness, we omit the chain index when there is no need to specify it.

Here $\nabla U(\boldsymbol{\theta})$ is the gradient of the energy function evaluated at $\boldsymbol{\theta}$, and $Z_{\Theta}(\boldsymbol{\theta})$ normalizes the distribution:

$$Z_{\Theta}(\boldsymbol{\theta}_i) = \sum_{\boldsymbol{\theta}_{i+1} \in \Theta} \exp\left(-\frac{1}{2\alpha}\left\|\boldsymbol{\theta}_{i+1} - \boldsymbol{\theta}_i - \frac{\alpha}{2\tau}\nabla U(\boldsymbol{\theta}_i)\right\|_2^2\right). \tag{6}$$

This proposal distribution allows DLS to make larger, parallel updates while maintaining computational efficiency. As the dimension $\mathbf{d}$ in parameter space grows, the cost of computing (6) becomes prohibitively expensive. A key insight is that the update rule can be factorized by coordinate:

$$\theta_{i+1,d} \sim \text{Categorical}\left[\text{Softmax}\left(\frac{1}{2\tau}\nabla U(\boldsymbol{\theta}_i)_d(\theta_{i+1,d} - \theta_{i,d}) - \frac{(\theta_{i+1,d} - \theta_{i,d})^2}{2\alpha}\right)\right], \tag{7}$$

where $d = 1, 2, \ldots, \mathbf{d}$ is the dimension index, and $\theta_{i,d}$ represents the $i$-th sample in the $d$-th dimension. This algebraic expansion, following the binomial theorem, works because $(\nabla U(\boldsymbol{\theta}_i)_d)^2$ is independent of $\theta_{i+1,d}$. It makes DLS scalable and computationally efficient for complex distributions (Zhang et al., 2022b). Furthermore, the first term in (7) biases the proposal towards low-energy regions, where the gradient points towards increasing probability; the second term acts as a regularizing factor, which penalizes large jumps unless they are strongly favored by the gradient.

DLS can operate with or without MH corrections. Without corrections, it is simplified to the Discrete Unadjusted Langevin Algorithm (DULA), which is computationally efficient but may introduce bias. With corrections in (4), it becomes the Discrete Metropolis-adjusted Langevin algorithm (DMALA), which corrects bias at an increased computational cost. Both DULA and DMALA employ non-local proposals specifically for the heat kernel to enable more efficient sampling (Sun et al., 2023a).

## 4 DISCRETE LANGEVIN SAMPLER WITH REPLICA EXCHANGE

The proposed DLS variants are present here, which incorporate replica exchange and a customized sampler swap mechanism to ensure detailed balance. The complete algorithm is provided at the end.

### 4.1 DISCRETE SAMPLERS WITH DIFFERENT TEMPERATURES

A key challenge with the naïve DLS is the tendency to become trapped in local modes, particularly in non-convex landscapes. To mitigate this, we introduce DREXEL, which incorporates replica exchange to enable efficient exploration across different local modes. Specifically, we employ two samplers separately with distinct step sizes and temperatures to approximate the target distribution:

$$\text{Categorical}\left[\text{Softmax}\left(\frac{1}{2\tau_k}\nabla U(\boldsymbol{\theta}_i^{(k)})_d\left(\theta_{i+1,d}^{(k)} - \theta_{i,d}^{(k)}\right) - \frac{\left(\theta_{i+1,d}^{(k)} - \theta_{i,d}^{(k)}\right)^2}{2\alpha_k}\right)\right], \quad k = 1, 2. \tag{8}$$

Here $\tau_1 < \tau_2$ and $\alpha_1 < \alpha_2$, with $k = 1$ being the low-temperature and $k = 2$ the high-temperature sampler. Intuitively, larger step sizes and higher temperatures encourage more exploratory moves, which allows the sampler to escape local modes through non-local jumps and explore different regions of the energy landscapes. This, on the downside, raises the rejection rate, as large jumps often land in low-probability regions, and introduce additional bias when approximating the target distribution.

To mitigate the bias, we further propose DREAM, which incorporates MH steps post-generation of new samples. Once the new samples are produced through (8), the acceptance rates $\mathcal{A}(\boldsymbol{\theta}_{i+1}^{(1)}, \boldsymbol{\theta}_i^{(1)})$ and $\mathcal{A}(\boldsymbol{\theta}_{i+1}^{(2)}, \boldsymbol{\theta}_i^{(2)})$ are estimated with (4). The new samples are accepted with probability $\mathcal{A}$ or rejected with $1 - \mathcal{A}$. The acceptance rates of two samplers are independent of one another. While the high-temperature sampler typically exhibits a lower acceptance rate than the low-temperature one, the rejection mechanism ensures that both samplers in DREAM converge to the target asymptotically.

It should be noted that while decaying step sizes are commonly advantageous in Langevin MCMC for handling big data (Teh et al., 2016), they present potential challenges in discrete sampling. In discrete spaces, small steps do not equate to gradual movements as they do in continuous spaces. Instead, they tend to repeatedly propose nearly identical samples, which causes the sampler to become trapped in local regions. This problem becomes severe when dealing with non-convex energy landscapes, where a decaying step size worsens the issue of local traps. For this reason, the MH step is often favored as

a solution in discrete sampling. With the MH step and fixed step sizes, the sampler can make large jumps to facilitate global exploration. This feature is essential for navigating highly structured state spaces, where the sampler needs flexibility to move between distant states.

In practice, high-temperature samplers may have difficulty exploiting certain regions due to abrupt exploration, which requires excessive time to fully characterize local modes and achieve mixing.

### 4.2 Sample Swaps between Discrete Samplers

A typical solution is to implement a swap function that enables sample exchanges between samplers at different temperatures. This helps cross energy barriers by combining the exploration of high-temperature samplers with the exploitation of low-temperature ones, which improves mixing rates.

The naïve swap function (3) of reMCMC relies on energy calculations at the current samples and corresponding temperatures. However, it is not practical to handle large-scale data in mini-batch settings. Intuitively, while $\tilde{U}(\boldsymbol{\theta}_{i+1}^{(1)})$ and $\tilde{U}(\boldsymbol{\theta}_{i+1}^{(2)})$ are both unbiased in mini-batches, a non-linear transformation of these estimators fail to provide an unbiased estimator for $S(\boldsymbol{\theta}_{i+1}^{(1)}, \boldsymbol{\theta}_{i+1}^{(2)})$ (Deng et al., 2020). Under normality assumption for the energy estimate, we consider a bias correction term:

$$\tilde{S}(\boldsymbol{\theta}_{i+1}^{(1)}, \boldsymbol{\theta}_{i+1}^{(2)}) = e^{\left(\frac{1}{\tau_2} - \frac{1}{\tau_1}\right)\left[U(\boldsymbol{\theta}_{i+1}^{(1)}) - U(\boldsymbol{\theta}_{i+1}^{(2)}) + \left(\frac{1}{\tau_1} - \frac{1}{\tau_2}\right)\sigma^2\right]}, \tag{9}$$

where $\sigma^2$ compensates for noise in the stochastic gradient and removes swap bias. This adjustment ensures that the swap function behaves as a Martingale and matches the expected value obtained from exact gradients. Although this correction is not strictly necessary in discrete sampling, we retain this design in practice and examine the potential need for bias correction in the experiments. The bias-corrected versions of DREXEL and DREAM are referred to as bDREXEL and bDREAM.

When reMCMC is applied to discrete spaces, a notable challenge arises: the decaying step sizes commonly employed in continuous settings are not applicable. To ensure asymptotic convergence to the target distribution with fixed step sizes, we must maintain detailed balance not only between the low-temperature and high-temperature samplers but also between the current and next output samples. The swap designs in (3) and (9), however, overlook energy and temperature differences. This potentially violates detailed balance and slows down mixing in discrete sampling tasks. To mitigate the imbalance, we propose a swap function tailored for discrete sampling:

$$\tilde{S}(\boldsymbol{\theta}_{i+1}^{(1)}, \boldsymbol{\theta}_{i+1}^{(2)} \mid \boldsymbol{\theta}_i^{(1)}, \boldsymbol{\theta}_i^{(2)}) = e^{\left(\frac{1}{\tau_2} - \frac{1}{\tau_1}\right)\left[U(\boldsymbol{\theta}_{i+1}^{(1)}) + U(\boldsymbol{\theta}_i^{(1)}) - U(\boldsymbol{\theta}_{i+1}^{(2)}) - U(\boldsymbol{\theta}_i^{(2)})\right]}. \tag{10}$$

This swap function incorporates energy estimates at the last samples, which respects the energy landscape and preserves detailed balance. Importantly, since the previous samples are treated as constants during the swap, the detailed balance between replicas remains unaffected. We will demonstrate how this correction guarantees asymptotic convergence to the target distribution in the next section.

### 4.3 The Proposed Algorithms

As outlined in Algorithm 1, we present DREXEL and DREAM for discrete sampling. The approaches employ two DLSs with distinct temperatures and step sizes, which allows for sample swaps between them. At each iteration, the current samples are updated, followed by MH steps in DREAM. The swap mechanism exchanges samples when the high-temperature sampler locates a lower-energy mode. After $I$ iterations, the low-temperature sampler outputs samples to characterize the energy landscape. This approach, discussed further in 5.1, improves the mixing rate over DLSs by balancing exploration and exploitation.

---

**Algorithm 1** DREXEL and DREAM.

---

**Input** Step Sizes $\alpha_1, \alpha_2$
**Input** Temperatures $\tau_1, \tau_2$.
**Input** Swap Intensity $\rho > 0$.
**Input** Initial Samples $\boldsymbol{\theta}_0^{(k)} \in \Theta$, $k = 1, 2$.
 1: **For** $i = 1, 2, \cdots, I$ **do**
 2:    **Sampling Steps:**
 3:    **For** $k = 0, 1, 2$ **do:**
 4:      **For** $d = 1, 2, \cdots, \mathbf{d}$ **do:**
 5:        Construct $q_d^{(k)}(\boldsymbol{\theta}^{(k)} \mid \boldsymbol{\theta}_i^{(k)})$ following (8)
 6:        Sample $\omega_d^{(k)} \sim q_d^{(k)}(\cdot \mid \boldsymbol{\theta}_i^{(k)})$
 7:      **End For**
 8:    **End For**
 9:    **MH Steps (for DREAM):**
10:    **For** $k = 1, 2$ **do:**
11:      Compute $\mathcal{A}(\boldsymbol{\theta}_i^{(k)}, \boldsymbol{\theta}_{i+1}^{(k)})$ following (4)
12:      Generate a number $u \sim U[0, 1]$
13:      Set $\boldsymbol{\theta}_{i+1}^{(k)} \leftarrow \omega^{(k)}$ if $u \leq \mathcal{A}$ else $\boldsymbol{\theta}_{i+1}^{(k)} \leftarrow \boldsymbol{\theta}_i^{(k)}$
14:    **End For**
15:    **Swapping Steps:**
16:    Generate a number $u \sim U[0, 1]$.
17:    Compute $\tilde{S}(\boldsymbol{\theta}_{i+1}^{(1)}, \boldsymbol{\theta}_{i+1}^{(2)})$ following (10)
18:    Swap $\boldsymbol{\theta}_{i+1}^{(1)}$ and $\boldsymbol{\theta}_{i+1}^{(2)}$ if $u \leq \rho \min\left\{1, \tilde{S}\right\}$
19: **End For**
**Output** Samples $\{\boldsymbol{\theta}_i^{(1)}\}_{i=1}^I$.

---

## 5   Theoretical Analysis

In the previous section, we introduced DREXEL and DREAM, which use factorization to allow parallel updates and employ swap mechanisms to improve non-convex exploration. While these features are beneficial, the overall performance heavily relies on their convergence properties and theoretical guarantees. In this section, we provide asymptotic convergence guarantees for DREXEL (i.e. the version without the MH correction).

### 5.1   Asymptotic Convergence on Log-Quadratic Distributions

Our focus is first on the asymptotic behaviors of DREXEL. The analysis aims to show that as step sizes approach zero, the algorithm exhibits zero asymptotic bias, which ensures accurate sampling from the target distribution. Specifically, we focus on log-quadratic energy $\pi(\boldsymbol{\theta}) \propto \exp\left(\boldsymbol{\theta}^\top \boldsymbol{J} \boldsymbol{\theta} + \boldsymbol{b}^\top \boldsymbol{\theta}\right)$, where $\boldsymbol{J} \in \mathbb{R}^{\mathbf{d} \times \mathbf{d}}$ is a symmetric matrix, and $\boldsymbol{b} \in \mathbb{R}^{\mathbf{d}}$ is a vector. If $\boldsymbol{J}$ is asymmetric, we apply spectral decomposition to obtain a symmetric matrix, which enables an analytically tractable solution.

Zhang et al. (2022b) showed that DLS with temperature 1 is reversible for log-quadratic energy distributions when the step size is sufficiently small. However, this result does not directly extend to the proposed algorithm, as the swap mechanism (3) introduces potential imbalances. This imbalance is due to discontinuous transitions between neighboring states in discrete space, which makes the swap acceptance rule insufficient to maintain a detailed balance. This further introduces bias during the sampling process and leads to inaccurate modeling. To address this, we carefully control the swap probability in (10) to regulate transitions between high- and low-temperature samplers.

**Theorem 1.** *Let $\alpha_1$ and $\alpha_2$ be the step sizes for the low- and high-temperature samplers, and let $q(\cdot|\boldsymbol{\theta})$ be the Markov chain transition kernel. Suppose the target $\pi(\boldsymbol{\theta})$ is log-quadratic, then:*

- *The Markov chain induced by DREXEL is reversible with respect to an intermediate distribution $\tilde{\pi}$, i.e., for all $\boldsymbol{\theta}, \boldsymbol{\theta}' \in \Theta$, $\tilde{\pi}(\boldsymbol{\theta})q(\boldsymbol{\theta}'|\boldsymbol{\theta}) = \tilde{\pi}(\boldsymbol{\theta}')q(\boldsymbol{\theta}|\boldsymbol{\theta}')$.*
- *As $\alpha_1, \alpha_2 \to 0$, the stationary distribution $\pi'$ converges weakly to the target distribution $\pi$.*

This analysis focuses on the state transition of the low-temperature sampler, as the high-temperature sampler only facilitates exploration and does not produce final samples. Intuitively, with probability $\rho\tilde{S}$, the next low-temperature sample is drawn from the high-temperature sampler, and with probability $\rho(1-\tilde{S})$, it selects from the low-temperature sampler. The transition simplifies to DLS without swaps and directly maintains the detailed balance, but the swap probability becomes essential for preserving this balance once swaps are considered in discrete sampling. Our designed swap function ensures that the overall transition dynamics remain balanced, as demonstrated in Appendix D.1.

## 6 EXPERIMENTS

To illustrate the effectiveness of our approach, we evaluate the proposed samplers across distinct discrete sampling and generative tasks. Our approach is compared against baselines including DLS (DULA and DMALA from Zhang et al. (2022b)), Any-scale Balanced sampling (AB) (Sun et al., 2023a), and the Automatic Cyclical Sampler (ACS) (Pynadath et al., 2024). More details such as experimental setups, hyper-parameters, and additional experimental results can refer to Appendix E.

### 6.1 SAMPLING FROM 2D SYNTHETIC PROBLEMS

We explore the challenges of sampling from 2D discrete multi-modal distributions defined over the domain $\Theta = \{1, 2, \ldots, N\}^{\mathbf{d}}$, where $N = 256$ and $\mathbf{d} = 101 \times 101$. Each coordinate takes one of the discrete values. Figure 2 (top) highlights the challenges of approximating non-convex energy landscapes, where samplers often struggle to explore the landscapes effectively with limited samples.

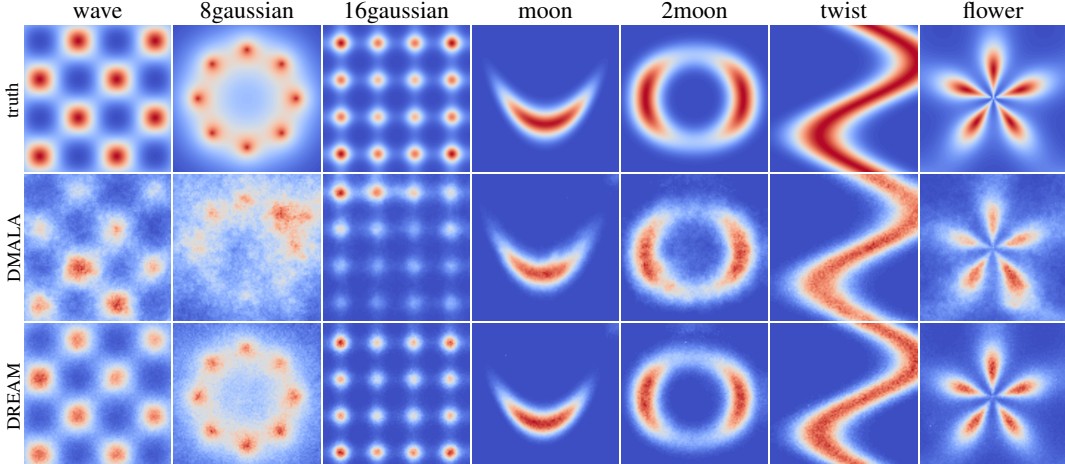

Figure 2: **Top:** Visualization of the true energy landscapes. **Middle:** Empirical energy distributions from DMALA. **Bottom:** Empirical energy distributions from DREAM.

Table 1: Experiment results with exploring 2D synthetic distributions, recorded with KL and MMD.

| Metric | Sampler | wave | 8gaussian | 16gaussian | moon | 2moon | twist | flower |
|---|---|---|---|---|---|---|---|---|
| KL($10^{-2}$) ↓ | DMALA | 2.419 | 1.337 | 7.690 | 2.397 | 4.848 | 3.767 | 2.765 |
| | AB | 1.028 | 0.851 | 3.373 | 2.567 | **4.127** | 3.033 | 2.421 |
| | ACS | 0.930 | 0.521 | 3.145 | 2.059 | 4.207 | 2.154 | 2.479 |
| | **DREAM** | **0.914** | **0.519** | **3.017** | **1.652** | 4.252 | **2.145** | **2.277** |
| MMD↓ | DMALA | 2.085 | 2.084 | 1.977 | 2.095 | 2.019 | 2.049 | 2.100 |
| | AB | 2.036 | 2.057 | **1.910** | 1.911 | 2.070 | 2.059 | 1.983 |
| | ACS | 2.014 | 2.028 | 1.984 | 1.996 | 2.068 | 2.047 | 1.966 |
| | **DREAM** | **1.969** | **2.007** | 1.922 | **1.908** | **2.014** | **1.976** | **1.913** |

DREAM is compared with DMALA, AB, and ACS in these tasks. With automatic differentiation for gradient computation, we generate 640,000 samples to form the empirical distributions. Figure 2 (bottom) provides a qualitative analysis, showing that DREAM can effectively capture the underlying complex distributions. In the wave, 8gaussians, and 16gaussians, DMALA (mid) captures only 50%

modes due to its tendency to get stuck in local minima. DREAM, by contrast, recovers all modes, which reflects its robust exploration across different tasks. For quantitative evaluation, we report Kullback–Leibler (KL) divergence and maximum mean discrepancy (MMD) as performance metrics (Blessing et al., 2024). As shown in Table 1, DREAM consistently outperforms the baselines across all distributions.

## 6.2 SAMPLING FROM ISING MODELS

The Ising model (Newman & Barkema, 1999) is a mathematical structure used to describe systems of interacting binary variables, which are commonly represented as spins in physical systems (MacKay, 2003). Each spin can assume binary states and interact with adjacent spins within a lattice. The interactions between these neighboring spins are governed by the energy function $U(\boldsymbol{\theta}) = w\boldsymbol{\theta}^{\top}\boldsymbol{J}\boldsymbol{\theta} + \boldsymbol{b}^{\top}\boldsymbol{\theta}$, where $\boldsymbol{\theta} \in \{-1, 1\}^{\mathbf{d}}$ is binary random variable, $\boldsymbol{J} \in \{0, 1\}^{\mathbf{d}\times\mathbf{d}}$ is a binary adjacency matrix, $w \in \mathbb{R}^{+}$ denote the connectivity strength, and $\boldsymbol{b} \in \{0, 1\}^{\mathbf{d}}$ is the bias vector.

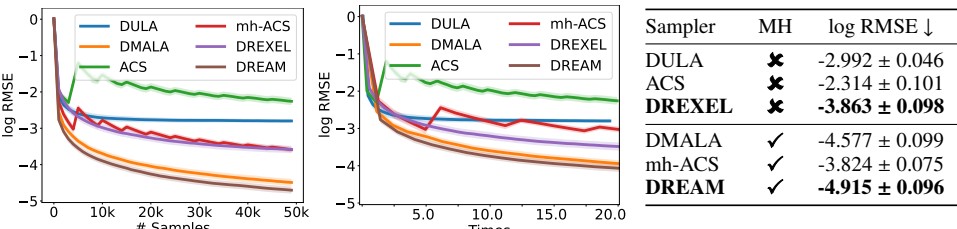

| Sampler | MH | log RMSE ↓ |
|---|---|---|
| DULA | ✘ | -2.992 ± 0.046 |
| ACS | ✘ | -2.314 ± 0.101 |
| **DREXEL** | ✘ | **-3.863 ± 0.098** |
| DMALA | ✓ | -4.577 ± 0.099 |
| mh-ACS | ✓ | -3.824 ± 0.075 |
| **DREAM** | ✓ | **-4.915 ± 0.096** |

Figure 3: Ising model sampling results, evaluated by log RMSE. DREAM yields the best scores.

From Figure 3, for samplers without MH corrections, DREXEL shows a fast and consistent reduction in log RMSE, while DULA and ACS converge more slowly. Note that ACS exhibits periodic fluctuations due to its cyclical step size, where error initially decreases but then increases as the step size decays. For discrete sampling, it implies small step sizes do not effectively exploit local modes, and decaying step sizes may not be as effective as MH corrections in discrete settings. Among samplers with MH corrections, DREAM delivers the most efficient error reduction, which benefits from a strong exploration-exploitation balance. DMALA follows a more gradual path, while mh-ACS mirrors the periodic behavior of ACS due to its similar step-size schedule. These findings indicate that the proposed samplers generally offer better and more reliable mixing rates.

## 6.3 SAMPLING FROM RESTRICTED BOLTZMANN MACHINES

Restricted Boltzmann Machines (RBMs) are generative stochastic neural networks designed to model complex distributions over discrete data (Fischer & Igel, 2012). RBMs typically consist of binary-valued hidden and visible units, where the visible units represent observed data and the hidden units capture latent dependencies in the data. The energy function $U(\boldsymbol{\theta}) = \log\left[1 + \exp\left(\boldsymbol{J}\boldsymbol{\theta} + \boldsymbol{c}\right)\right] + \boldsymbol{b}^{\top}\boldsymbol{\theta}$, where $\boldsymbol{\theta} \in \{0, 1\}^{\mathbf{d}}$ represents the binary state vector for the visible layer, $\boldsymbol{J} \in \mathbb{R}^{\mathbf{m}\times\mathbf{d}}$ is the weight matrix, $\boldsymbol{c} \in \mathbb{R}^{\mathbf{m}}$ and $\boldsymbol{b} \in \mathbb{R}^{\mathbf{d}}$ denote biases for hidden units and visible units correspondingly.

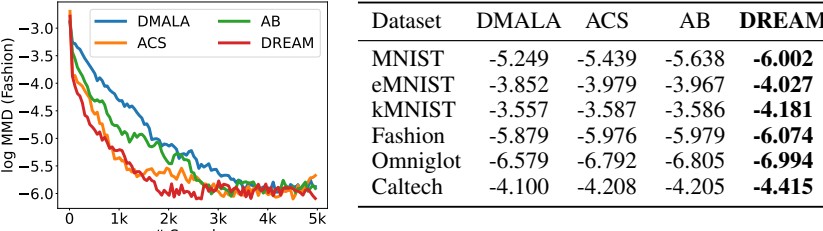

| Dataset | DMALA | ACS | AB | **DREAM** |
|---|---|---|---|---|
| MNIST | -5.249 | -5.439 | -5.638 | **-6.002** |
| eMNIST | -3.852 | -3.979 | -3.967 | **-4.027** |
| kMNIST | -3.557 | -3.587 | -3.586 | **-4.181** |
| Fashion | -5.879 | -5.976 | -5.979 | **-6.074** |
| Omniglot | -6.579 | -6.792 | -6.805 | **-6.994** |
| Caltech | -4.100 | -4.208 | -4.205 | **-4.415** |

Figure 4: RBM sampling results, quantified by MMD. DREAM outperforms across various datasets.

We trained RBMs with 500 hidden units on the MNIST dataset using contrastive divergence (Hinton, 2002). To benchmark the effectiveness of various samplers, we reported MMDs between outputs generated by each sampler and the one by the structured Block-Gibbs sampler specific to RBMs.

Figure 4 highlights DREAM as the most effective sampler, which consistently achieves the best log MMD across all datasets and suggests superior convergence in sampling from RBMs. AB and ACS perform nearly as well but with slightly higher MMD values. Overall, DREAM provides the most robust sampling, followed by AB and ACS, while DMALA trails behind.

## 6.4 LEARNING ENERGY-BASED MODELS

Deep Energy-Based Models (EBMs) (Ngiam et al., 2011; Bond-Taylor et al., 2021) are a class of probabilistic models where the energy function is parameterized by a ResNet (He et al., 2016). Specifically, the probability of a data point $x$ is given by $P_\theta(x) = \exp\left[E_\theta(x)\right]/Z_\theta$, where $E_\theta(x)$ is the energy function parameterized by $\theta$, and $Z_\theta = \mathbb{E}_{\theta \sim \Theta} \exp\left[E_\theta(x)\right]$ normalizes the distribution.

With DULA and DMALA as baselines, we evaluate DREXEL and DREAM[2] by learning Deep EBMs. During training, the intractable likelihood gradient of the model is approximated through Persistent Contrastive Divergence (Tieleman & Hinton, 2009), while a replay buffer (Du & Mordatch, 2019) containing 1,000 past samples is implemented to improve both the efficiency and stability of the training process. Each sampler runs for 40 steps per iteration. Upon completing training, Annealed Importance Sampling (Neal, 2001) is conducted with DULA to estimate the test log-likelihoods.

Table 2: Test log-likelihoods of Deep EBMs evaluated on image datasets.

| Dataset | DULA | DMALA | bDREXEL | bDREAM | DREXEL | **DREAM** |
|---|---|---|---|---|---|---|
| Static MNIST | -84.579 | -85.145 | -85.638 | -84.823 | -84.509 | **-83.929** |
| Dynamic MNIST | -86.625 | -84.799 | -86.907 | -85.104 | -83.984 | **-82.963** |
| Omniglot | -118.541 | -111.820 | -102.405 | -100.042 | -101.930 | **-98.454** |
| Caltech | -108.626 | -107.820 | -108.199 | -107.899 | -93.481 | **-92.003** |

We trained Deep EBMs for 20,000 iterations on binary images from Static MNIST, Dynamic MNIST, Omniglot, and Caltech Silhouettes datasets. The test log-likelihoods for trained models across different samplers are recorded in Table 2. Among the samplers, DREAM consistently achieved the highest log-likelihoods across all datasets, with notable improvements on Omniglot and Caltech Silhouettes. For MNIST datasets, DREXEL and DREAM also showed competitive performance, particularly on Static MNIST. In contrast, bDREXEL and bDREAM generally performed worse, with DREXEL and DREAM showing clear superiority across most datasets. These findings confirm that MH steps are essential for improving performance in discrete sampling tasks. Also, the proposed swap mechanism in (10) is effective at correcting imbalance and yielding better log-likelihood estimates across diverse image datasets.

## 7 CONCLUSION AND DISCUSSION

In this work, we addressed the challenge of balancing global exploration and local exploitation in non-convex discrete energy landscapes by proposing DREXEL and DREAM. These samplers integrate DLS with replica exchange to overcome the limitations of traditional samplers, which tend to get trapped in local modes due to reliance on local gradients and small disturbances.

We theoretically prove that the proposed samplers are reversible, which guarantees the accurate preservation of the target distribution. Moreover, these samplers achieve faster mixing than the naïve DLS. The empirical evidence suggests that the proposed samplers and swap mechanism significantly improve exploration and mixing in non-convex discrete spaces. Furthermore, while DREXEL maintains detailed balance throughout the process, MH corrections are critical for optimizing performance in certain tasks.

Our current work focuses on designing a single low-temperature and a single high-temperature sampler. Future research could extend this framework by introducing multiple parallel samplers to enhance exploration. We will also study theoretical guarantees to quantify the acceleration effect of the swap mechanism in the future.

---

[2]Unless stated otherwise, DREXEL and DREAM refer to samplers based on the swap design from (10). bDREXEL and bDREAM apply the swap function described in (9).

ETHICS STATEMENT

We adhere to the ICLR Code of Ethics and confirm that our experiments use only public datasets. While our results are primarily based on standard benchmarks, we recognize the potential for misuse and encourage responsible application of our methods on real-world data.

REPRODUCIBILITY STATEMENT

To facilitate the reproducibility of our proposed method, the implementation of the synthetic task related to this work is made available at the following anonymous link: https://anonymous.4open.science/r/dream-F7E6.

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

## A  FUTHER DISCUSSION ON RELATED WORK

**Modeling Non-Convex Discrete Distributions** is important in modern machine learning tasks. Swendsen & Wang (1987) designed cluster updates in Monte Carlo simulations to efficiently navigate complex energy landscapes by updating groups of variables simultaneously. Wolff (1989) further improved sampling efficiency by flipping single large clusters of spins, thereby reducing autocorrelation times near critical points in non-convex distributions. Marinari & Parisi (1992) considered temperature as a dynamic variable and allowed the system to overcome energy barriers and explore multiple modes of a non-convex distribution effectively. While these models might work for specific discrete structures, they rely on random walk or Gibbs sampling. Their extensions to gradient-based discrete sampling may significantly improve their efficiency in non-convex exploration.

**Related Discrete Methods** have been made in modeling and optimizing discrete distributions. Zhang et al. (2022a) introduced energy-based generative flow networks to amortize expensive MCMC exploration into a fixed number of actions. Discrete Diffusion Models (Sun et al., 2023b; Lou et al., 2024) extended continuous-time diffusion models to discrete spaces with well-defined score functions for discrete variables. Wen et al. (2024) proposed an efficient gradient-based discrete optimization method for generative models. These approaches to discretizing continuous methods and handling complex discrete data offer valuable insights for developing DLSs.

**Stochastic Gradient Langevin MCMC** (Welling & Teh, 2011) has become a favored MCMC method in big data due to its effective transition from optimization to sampling. However, its lack of adaptive step sizes to the energy curvature limits the use of this crucial information. To further leverage curvature information, Quasi-Newton methods (Ahn et al., 2012; Simsekli et al., 2016) exploit curvature information by adjusting step sizes, while Hamiltonian Monte Carlo (Neal, 2012; Campbell et al., 2021) and higher-order approaches (Chen et al., 2015; Li et al., 2019) employ larger step sizes to improve stability. These approaches, however, still encounter difficulties in avoiding local traps, which is where advanced reMCMC methods help balance exploration and exploitation when navigating non-convex energy landscapes.

## B  DREXEL AND DREAM WITH BINARY VARIABLES

When the variable domain $\Theta$ is binary $\{0, 1\}^{\mathbf{d}}$, Algorithm 1 can be further simplified. In this binary setting, the Hadamard product, denoted by $\odot$, simplifies several operations. This streamlined version demonstrates that both DREXEL and DREAM can be efficiently parallelized across CPUs and GPUs, which leads to reduced computational cost.

Binary variables are particularly advantageous in this context. The binary domain facilitates the use of efficient bitwise operations, which not only speed up the computation but also enable the algorithm to scale better in high-dimensional spaces. Moreover, the simplicity of the binary domain reduces algorithmic complexity, which can facilitate computational efficiency.

## C  DREXEL AND DREAM WITH CATEGORICAL VARIABLES

We further examine how DREXEL and DREAM can be formulated for categorical variables using one-hot vectors and ordinal integers.

In **one-hot encoding**, each categorical variable $\boldsymbol{\theta}_i$ is represented as a vector in $\{0, 1\}^N$ where exactly one element is 1, and the rest are 0. The update rule for one-hot encoded variables is given by:

$$
\text{Categorical}\left[\text{Softmax}\left(\frac{1}{2\tau_k}\nabla U(\boldsymbol{\theta}_i^{(k)})_d\left(\theta_{i+1,d}^{(k)} - \theta_{i,d}^{(k)}\right) - \frac{\left\|\theta_{i+1,d}^{(k)} - \theta_{i,d}^{(k)}\right\|_2^2}{2\alpha_k}\right)\right], \quad k = 1, 2.
$$

In this setting, the difference $\boldsymbol{\theta}_{i+1}^{(k)} - \boldsymbol{\theta}_i^{(k)}$ results in a vector with exactly two non-zero elements, which reflects a transition between categories.

For **ordinal variables**, where categories have a natural ordering, $\boldsymbol{\theta}_i$ can be represented as integers in $\{0, 1, \cdots, N-1\}$. The update rule becomes:

$$\text{Categorical}\left[\text{Softmax}\left(\frac{1}{2\tau_k}\nabla U(\boldsymbol{\theta}_i^{(k)})_d\left(\theta_{i+1,d}^{(k)}-\theta_{i,d}^{(k)}\right)-\frac{\left(\theta_{i+1,d}^{(k)}-\theta_{i,d}^{(k)}\right)^2}{2\alpha_k}\right)\right], \quad k=1,2.$$

Here, the scalar difference $(\theta_{i+1,d}^{(k)}-\theta_{i,d}^{(k)})$ captures the magnitude and direction of the transition between ordered categories. This representation leverages the ordering information to inform the proposal distribution more precisely.

---

**Algorithm 2** DREXEL or DREAM with Binary Variables.

---

**Input** Step Sizes $\alpha_1, \alpha_2$, Temperatures $\tau_1, \tau_2$, and Swap Intensity $\rho > 0$.
**Input** Initial Samples $\boldsymbol{\theta}_0^{(k)} \in \Theta$, $k = 1, 2$.

1: **For** $i = 1, 2, \cdots, I$ **do**
2:    **Sampling Steps:**
3:    **For** $k = 0, 1, 2$ **do:**
4:       Compute $P_k(\boldsymbol{\theta}_i^{(k)}) = \dfrac{\exp\left(-\frac{1}{2\tau_k}\nabla U(\boldsymbol{\theta}_i^{(k)})\odot(2\boldsymbol{\theta}_i^{(k)}-1)-\frac{1}{2\alpha_k\tau_k}\right)}{\exp\left(-\frac{1}{2\tau_k}\nabla U(\boldsymbol{\theta}_i^{(k)})\odot(2\boldsymbol{\theta}_i^{(k)}-1)-\frac{1}{2\alpha_k\tau_k}\right)+1}$
5:       Sample $\boldsymbol{u} \sim U([0,1]^{\mathbf{d}})$
6:       Set $\boldsymbol{I}_k \leftarrow \text{dim}(\boldsymbol{u} \le P_k(\boldsymbol{\theta}_i^{(k)}))$
7:       Set $\boldsymbol{\omega}^{(k)} \leftarrow \text{flipdim}(I_k)$
8:    **End For**
9:    **MH Steps (for DREAM):**
10:    **For** $k = 1, 2$ **do:**
11:       Compute $q_k(\boldsymbol{\omega}^{(k)}|\boldsymbol{\theta}_i^{(k)}) = \Pi_{d\in\boldsymbol{I}_k}\mathbb{P}(\boldsymbol{\theta}_i^{(k)})_d \cdot \Pi_{d\notin\boldsymbol{I}_k}\left(1-\mathbb{P}(\boldsymbol{\theta}_i^{(k)})_d\right)$
12:       Compute $q_k(\boldsymbol{\theta}_i^{(k)}|\boldsymbol{\omega}^{(k)}) = \Pi_{d\in\boldsymbol{I}_k}\mathbb{P}(\boldsymbol{\omega}^{(k)})_d \cdot \Pi_{d\notin\boldsymbol{I}_k}\left(1-\mathbb{P}(\boldsymbol{\omega}^{(k)})_d\right)$
13:       Compute $P_k(\boldsymbol{\omega}^{(k)}) = \dfrac{\exp\left(-\frac{1}{2\tau_k}\nabla U(\boldsymbol{\omega}^{(k)})\odot(2\boldsymbol{\omega}^{(k)}-1)-\frac{1}{2\alpha_k\tau_k}\right)}{\exp\left(-\frac{1}{2\tau_k}\nabla U(\boldsymbol{\omega}^{(k)})\odot(2\boldsymbol{\omega}^{(k)}-1)-\frac{1}{2\alpha_k\tau_k}\right)+1}$
14:       Compute $\mathcal{A}(\boldsymbol{\omega}^{(k)},\boldsymbol{\theta}_i^{(k)})$ follows (4)
15:       Generate a number $u \sim U[0,1]$
16:       Set $\boldsymbol{\theta}_{i+1}^{(k)} \leftarrow \boldsymbol{\omega}^{(k)}$ if $u \le \mathcal{A}$ else $\boldsymbol{\theta}_{i+1}^{(k)} \leftarrow \boldsymbol{\theta}_i^{(k)}$
17:    **End For**
18:    **Swapping Steps:**
19:    Generate a number $u \sim U[0,1]$.
20:    Compute $\tilde{S}(\boldsymbol{\theta}_{i+1}^{(1)},\boldsymbol{\theta}_{i+1}^{(2)})$ follows (10)
21:    Swap $\boldsymbol{\theta}_{i+1}^{(1)}$ and $\boldsymbol{\theta}_{i+1}^{(2)}$ if $u \le \rho\min\left\{1,\tilde{S}\right\}$
22: **End For**
**Output** Samples $\{\boldsymbol{\theta}_i^{(1)}\}_{i=1}^{I}$.

---

# D   THEORETICAL ANALYSIS

In this section, we first validate the assumptions of smoothness, dissipativity, and the fine grid used in our analysis. Next, we provide an asymptotic analysis to show the weak convergence of DREXEL to the target distribution. We conclude with a detailed examination of its non-asymptotic behavior.

## D.1   PROOF OF THEOREM 1

To explore the reversibility of the discrete replica exchange Langevin sampler, we follow the proof of Zhang et al. (2022b). For the discrete replica exchange Langevin sampler, we denote the update rule as follows:

$$q_1\left(\boldsymbol{\theta}' \mid \boldsymbol{\theta}^{(1)}\right) \propto \text{Categorical}\left(\text{Softmax}\left(\frac{1}{\tau_1}\nabla U(\boldsymbol{\theta}^{(1)})_d(\theta'_d - \theta_d^{(1)}) - \frac{(\theta'_d - \theta_d^{(1)})^2}{2\alpha_1}\right)\right)$$

$$q_2\left(\boldsymbol{\theta}' \mid \boldsymbol{\theta}^{(2)}\right) \propto \text{Categorical}\left(\text{Softmax}\left(\frac{1}{\tau_2}\nabla U(\boldsymbol{\theta}^{(2)})_d(\theta'_d - \theta_d^{(2)}) - \frac{(\theta'_d - \theta_d^{(2)})^2}{2\alpha_2}\right)\right),$$

where $d = 1, 2, \cdots, \mathbf{d}$, $\forall \boldsymbol{\theta}^{(1)}, \boldsymbol{\theta}^{(2)} \in \Theta$, $\Theta \in \mathbb{R}^{\mathbf{d}}$. $\boldsymbol{\theta}^{(1)}$ denotes the current sample from the low-temperature sampler, and $\boldsymbol{\theta}^{(2)}$ is the sample from the high-temperature sampler.

*Proof.* We consider the transition probability $q\left(\boldsymbol{\theta}' \mid \boldsymbol{\theta}^{(1)}\right)$. Different from the direct transition in DLS, we consider two scenarios to transition from $\boldsymbol{\theta}^{(1)}$ to $\boldsymbol{\theta}'$: with probability $1 - S$, there is no chain swap, and the model parameter change from $\boldsymbol{\theta}^{(1)}$ to $\boldsymbol{\theta}'$ in the low-temperature sampler; with probability $S$, there is a chain swap, and the high-temperature sampler generate new sample from $\boldsymbol{\theta}^{(2)}$ to $\boldsymbol{\theta}'$. We recall the definition of the proposed swap function $S(\cdot, \cdot)$ in (10) as follows:

$$S\left(\boldsymbol{\theta}_{i+1}^{(1)}, \boldsymbol{\theta}_{i+1}^{(2)} \mid \boldsymbol{\theta}_i^{(1)}, \boldsymbol{\theta}_i^{(2)}\right) := e^{\left(\frac{1}{\tau_2} - \frac{1}{\tau_1}\right)\left[U\left(\theta_{i+1}^{(1)}\right) + U\left(\theta_i^{(1)}\right) - U\left(\theta_{i+1}^{(2)}\right) - U\left(\theta_i^{(2)}\right)\right]}.$$

Following this, we rewrite the transition probability $q\left(\boldsymbol{\theta}' \mid \boldsymbol{\theta}\right)$ of the discrete replica exchange Langevin sampler as:

$$q\left(\boldsymbol{\theta}' \mid \boldsymbol{\theta}_i^{(1)}\right) = \sum_{\boldsymbol{\theta}_i^{(2)}} \sum_{\boldsymbol{\theta}_{i+1}^{(2)}} \pi\left(\boldsymbol{\theta}_i^{(2)}\right) q_2\left(\boldsymbol{\theta}_{i+1}^{(2)} \mid \boldsymbol{\theta}_i^{(2)}\right)\left[1 - S\left(\boldsymbol{\theta}', \boldsymbol{\theta}_{i+1}^{(2)}\right)\right] q_1\left(\boldsymbol{\theta}' \mid \boldsymbol{\theta}_i^{(1)}\right)$$

$$+ \sum_{\boldsymbol{\theta}_i^{(2)}} \sum_{\boldsymbol{\theta}_{i+1}^{(1)}} \pi\left(\boldsymbol{\theta}_i^{(2)}\right) q_2\left(\boldsymbol{\theta}' \mid \boldsymbol{\theta}_i^{(2)}\right) S\left(\boldsymbol{\theta}_{i+1}^{(1)}, \boldsymbol{\theta}'\right) q_1\left(\boldsymbol{\theta}_{i+1}^{(1)} \mid \boldsymbol{\theta}_i^{(1)}\right), \tag{11}$$

where the first term on the right-hand side of (11) is the probability change from $\boldsymbol{\theta}$ to $\boldsymbol{\theta}'$ in the low-temperature sampler, and the second term is the probability that the low-temperature sampler starts with $\boldsymbol{\theta}_i^{(1)}$, the high-temperature chain ends on $\boldsymbol{\theta}'$, with a probability $S$ to have the sample swap.

To further demonstrate the reversibility of the proposed discrete replica exchange Langevin sampler, we multiply $\pi_\alpha(\theta)$ from both sides[3]:

$$\pi_\alpha\left(\boldsymbol{\theta}_i^{(1)}\right) q\left(\boldsymbol{\theta}' \mid \boldsymbol{\theta}_i^{(1)}\right)$$

$$= \sum_{\boldsymbol{\theta}_i^{(2)}} \sum_{\boldsymbol{\theta}_{i+1}^{(2)}} \pi_\alpha\left(\boldsymbol{\theta}_i^{(2)}\right) q_2\left(\boldsymbol{\theta}_{i+1}^{(2)} \mid \boldsymbol{\theta}_i^{(2)}\right)\left[1 - S\left(\boldsymbol{\theta}', \boldsymbol{\theta}_{i+1}^{(2)}\right)\right] \pi_\alpha\left(\boldsymbol{\theta}_i^{(1)}\right) q_1\left(\boldsymbol{\theta}' \mid \boldsymbol{\theta}_i^{(1)}\right)$$

$$+ \sum_{\boldsymbol{\theta}_i^{(2)}} \sum_{\boldsymbol{\theta}_{i+1}^{(1)}} \pi_\alpha\left(\boldsymbol{\theta}_i^{(2)}\right) q_2\left(\boldsymbol{\theta}' \mid \boldsymbol{\theta}_i^{(2)}\right) S\left(\boldsymbol{\theta}_{i+1}^{(1)}, \boldsymbol{\theta}'\right) \pi_\alpha\left(\boldsymbol{\theta}_i^{(1)}\right) q_1\left(\boldsymbol{\theta}_{i+1}^{(1)} \mid \boldsymbol{\theta}_i^{(1)}\right)$$

$$\overset{(a)}{=} \sum_{\boldsymbol{\theta}_i^{(2)}} \sum_{\boldsymbol{\theta}_{i+1}^{(2)}} \pi_\alpha\left(\boldsymbol{\theta}_i^{(2)}\right) q_2\left(\boldsymbol{\theta}_{i+1}^{(2)} \mid \boldsymbol{\theta}_i^{(2)}\right)\left[1 - e^{\left(\frac{1}{\tau_2} - \frac{1}{\tau_1}\right)\left[U(\theta') + U\left(\theta_i^{(1)}\right) - U\left(\theta_{i+1}^{(2)}\right) - U\left(\theta_i^{(2)}\right)\right]}\right] \pi_\alpha\left(\boldsymbol{\theta}_i^{(1)}\right) q_1\left(\boldsymbol{\theta}' \mid \boldsymbol{\theta}_i^{(1)}\right)$$

$$+ \sum_{\boldsymbol{\theta}_i^{(2)}} \sum_{\boldsymbol{\theta}_{i+1}^{(1)}} \pi_\alpha\left(\boldsymbol{\theta}_i^{(2)}\right) q_2\left(\boldsymbol{\theta}' \mid \boldsymbol{\theta}_i^{(2)}\right) \pi_\alpha\left(\boldsymbol{\theta}_i^{(1)}\right) q_1\left(\boldsymbol{\theta}_{i+1}^{(1)} \mid \boldsymbol{\theta}_i^{(1)}\right) e^{\left(\frac{1}{\tau_2} - \frac{1}{\tau_1}\right)\left[U\left(\theta_{i+1}^{(1)}\right) + U\left(\theta_i^{(1)}\right) - U(\theta') - U\left(\theta_i^{(2)}\right)\right]}, \tag{12}$$

---

[3]We ignore the subscript of the step size from $\alpha_1$ to $\alpha$ for the stationary distribution $\pi_\alpha$ for simplicity.

where (*a*) replace the swap function with (10).

Recall the assumption that the target distribution is defined as log-quadratic $\pi(\boldsymbol{\theta}) = \exp\left(\boldsymbol{\theta}^\top \boldsymbol{J}\boldsymbol{\theta} + \boldsymbol{b}^\top\boldsymbol{\theta}\right)/Z$, where $\boldsymbol{J} \in \mathbb{R}^{\mathbf{d}\times\mathbf{d}}$ is a symmetric matrix, $\boldsymbol{b} \in \mathbb{R}^{\mathbf{d}}$ is a vector, and $Z$ normalizes the distribution. We then have $\nabla U(\boldsymbol{\theta}) = 2\boldsymbol{J}^\top\boldsymbol{\theta} + \boldsymbol{b}$ and $\nabla^2 U(\boldsymbol{\theta}) = 2\boldsymbol{J}$ is a constant.

We further denote $Z_\alpha(\boldsymbol{\theta}) = \sum_{\boldsymbol{x}} \exp\left[\frac{1}{2}(U(\boldsymbol{x}) - U(\boldsymbol{\theta})) - (\boldsymbol{x}-\boldsymbol{\theta})^\top\left(\frac{1}{2\alpha}\mathbf{I} + \frac{1}{2}\boldsymbol{J}\right)(\boldsymbol{x}-\boldsymbol{\theta})\right]$ and $\pi_\alpha(\boldsymbol{\theta}) = Z_\alpha(\boldsymbol{\theta})\pi(\boldsymbol{\theta})/\sum_{\boldsymbol{x}} Z_\alpha(\boldsymbol{x})\pi(\boldsymbol{x})$. According to Theorem 1 in Zhang et al. (2022b), we have the transition from $\boldsymbol{\theta}_i^{(2)}$ to $\boldsymbol{\theta}_{i+1}^{(2)}$ multiplying the stationary distribution $\pi_\alpha\left(\boldsymbol{\theta}_i^{(2)}\right)$ as:

$$
\pi_\alpha\left(\boldsymbol{\theta}_i^{(2)}\right) q_1\left(\boldsymbol{\theta}_{i+1}^{(2)} \mid \boldsymbol{\theta}_i^{(2)}\right) = \frac{Z_\alpha\left(\boldsymbol{\theta}_i^{(2)}\right)\pi_\alpha\left(\boldsymbol{\theta}_i^{(2)}\right)}{\sum_{\boldsymbol{x}} Z(\boldsymbol{x})\pi_\alpha(\boldsymbol{x})} \cdot \frac{\exp\left[\frac{1}{2}\left(U\left(\boldsymbol{\theta}_{i+1}^{(2)}\right) - U(\boldsymbol{\theta}_i^{(2)})\right) - \left(\boldsymbol{\theta}_{i+1}^{(2)} - \boldsymbol{\theta}_i^{(2)}\right)^\top\left(\frac{1}{2\alpha}\mathbf{I} + \frac{1}{2}\boldsymbol{J}\right)\left(\boldsymbol{\theta}_{i+1}^{(2)} - \boldsymbol{\theta}_i^{(2)}\right)\right]}{Z_\alpha\left(\boldsymbol{\theta}_i^{(2)}\right)}
$$

$$
= \frac{\exp\left[\frac{1}{2}\left(U\left(\boldsymbol{\theta}_{i+1}^{(2)}\right) + U(\boldsymbol{\theta}_i^{(2)})\right) - \left(\boldsymbol{\theta}_{i+1}^{(2)} - \boldsymbol{\theta}_i^{(2)}\right)^\top\left(\frac{1}{2\alpha}\mathbf{I} + \frac{1}{2}\boldsymbol{J}\right)\left(\boldsymbol{\theta}_{i+1}^{(2)} - \boldsymbol{\theta}_i^{(2)}\right)\right]}{Z \cdot \sum_{\boldsymbol{x}} Z(\boldsymbol{x})\pi_\alpha(\boldsymbol{x})},
$$

which is symmetric. Similarly, we can expand the following distributions according to the definition of log-quadratic targets and the transition probabilities $\pi_\alpha\left(\boldsymbol{\theta}_i^{(1)}\right) q_1\left(\boldsymbol{\theta}' \mid \boldsymbol{\theta}_i^{(1)}\right)$, $\pi_\alpha\left(\boldsymbol{\theta}_i^{(2)}\right) q_2\left(\boldsymbol{\theta}' \mid \boldsymbol{\theta}_i^{(2)}\right)$, and $\pi_\alpha\left(\boldsymbol{\theta}_i^{(1)}\right) q_1\left(\boldsymbol{\theta}_{i+1}^{(1)} \mid \boldsymbol{\theta}_i^{(1)}\right)$. Similarly, all of them are symmetric, which indicates that $\pi_\alpha\left(\boldsymbol{\theta}_i^{(1)}\right) q\left(\boldsymbol{\theta}' \mid \boldsymbol{\theta}_i^{(1)}\right)$ is also symmetric. Therefore, we conclude that $q\left(\boldsymbol{\theta}' \mid \boldsymbol{\theta}_i^{(1)}\right)$ give in (12) is reversible and the stationary distribution is $\pi_\alpha\left(\boldsymbol{\theta}_i^{(1)}\right)$.

To further prove the stationary distribution $\pi_\alpha$ converges weakly to the target $\pi$ as the step sizes are close to zero, we first observe that for any $\boldsymbol{\theta} \in \Theta$,

$$
Z_\alpha(\boldsymbol{\theta}) = \sum_{\boldsymbol{x}\in\boldsymbol{\theta}} \exp\left(\frac{1}{2}\left(U(\boldsymbol{x}) - U(\boldsymbol{\theta})\right) - (\boldsymbol{x}-\boldsymbol{\theta})^\top\left(\frac{1}{2\alpha}\mathbf{I} + \frac{1}{2}\boldsymbol{J}\right)(\boldsymbol{x}-\boldsymbol{\theta})\right).
$$

As $\alpha \to 0$, the term involving $\frac{1}{2\alpha}\mathbf{I}$ dominates the quadratic form unless $\boldsymbol{x} = \boldsymbol{\theta}$. Specifically, for $\boldsymbol{x} \neq \boldsymbol{\theta}$,

$$
(\boldsymbol{x}-\boldsymbol{\theta})^\top\left(\frac{1}{2\alpha}\mathbf{I} + \frac{1}{2}\boldsymbol{J}\right)(\boldsymbol{x}-\boldsymbol{\theta}) \geq \frac{1}{2\alpha}\|\boldsymbol{x}-\boldsymbol{\theta}\|^2,
$$

which tends to infinity as $\alpha \to 0$. Therefore, the terms in the sum for $\boldsymbol{x} \neq \boldsymbol{\theta}$ become negligible, which means $\exp\left(-(\boldsymbol{x}-\boldsymbol{\theta})^\top\left(\frac{1}{2\alpha}\mathbf{I} + \frac{1}{2}\boldsymbol{J}\right)(\boldsymbol{x}-\boldsymbol{\theta})\right) \to 0$ as $\alpha \to 0$. For $\boldsymbol{x} = \boldsymbol{\theta}$, the exponent simplifies to zero: $\exp\left(\frac{1}{2}(U(\boldsymbol{\theta}) - U(\boldsymbol{\theta})) - 0\right) = 1$. Thus, we can conclude that $\lim_{\alpha\to0} Z_\alpha(\boldsymbol{\theta}) = 1$.

We denote the denominator of $\pi_\alpha(\boldsymbol{\theta})$ is $D_\alpha = \sum_{\boldsymbol{x}\in\Theta} Z_\alpha(\boldsymbol{x})\pi(\boldsymbol{x})$. Following the above derivation, we can easily find that $Z_\alpha(\boldsymbol{x}) \to 1$ as $\alpha \to 0$ for each $\boldsymbol{x} \in \Theta$. Therefore, we also have $\lim_{\alpha\to0} D_\alpha = \sum_{\boldsymbol{x}\in\boldsymbol{\theta}} \pi(\boldsymbol{x}) = 1$.

Combining the above results, we have $\lim_{\alpha\to0} \pi_\alpha(\boldsymbol{\theta}) = \lim_{\alpha\to0} \frac{Z_\alpha(\boldsymbol{\theta})\pi(\boldsymbol{\theta})}{D_\alpha} = \frac{\pi(\boldsymbol{\theta})}{1} = \pi(\boldsymbol{\theta})$. Thus, we derive a conclusion that $\pi_\alpha(\boldsymbol{\theta})$ converges point-wisely to $\pi(\boldsymbol{\theta})$ as $\alpha \to 0$.

Since $\pi_\alpha$ and $\pi$ are probability mass functions on a discrete space $\boldsymbol{\theta}$, and $\pi_\alpha(\boldsymbol{\theta}) \to \pi(\boldsymbol{\theta})$ point-wisely, according to Scheffé's Lemma (Billingsley, 2017), it further implies that $\lim_{\alpha\to0} \sum_{\boldsymbol{\theta}\in\Theta} |\pi_\alpha(\boldsymbol{\theta}) - \pi(\boldsymbol{\theta})| = 0$.

This convergence implies weak convergence of $\pi_\alpha$ to $\pi$: from Dominated Convergence Theorem (Folland, 1999), we have for any bounded function $f : \Theta \to \mathbb{R}$,

$$
\lim_{\alpha\to0} \sum_{\boldsymbol{\theta}\in\Theta} f(\boldsymbol{\theta})\pi_\alpha(\boldsymbol{\theta}) = \sum_{\boldsymbol{\theta}\in\boldsymbol{\theta}} f(\boldsymbol{\theta})\pi(\boldsymbol{\theta}).
$$

This completes the proof that $\pi_\alpha$ converges weakly to $\pi$ as $\alpha \to 0$.

$\square$

# E ADDITIONAL EXPERIMENTAL RESULTS

The experiments were run on a server featuring an Intel(R) Core(TM) i9-14900K processor, RTX 4090 GPUs, and 128 GB DDR4 memory.

### E.1 Sampling from 2D Synthetic Energies

To evaluate the efficiency of MCMC algorithms in exploring non-convex discrete energy landscapes, we consider a set of energy functions that present varying degrees of complexity and multimodality. These functions are designed to test the algorithms' ability to navigate challenging landscapes characterized by multiple local minima, sharp ridges, and disconnected modes.

**Wave** energy function is a periodic sinusoidal surface with alternating peaks and valleys:

$$U(x, y) = \sin(3x) \sin(3y).$$

Its highly rugged landscape arises from repeated oscillations, which makes it a suitable test for an algorithm's ability to navigate sharp and oscillating energy contours without getting trapped in local minima.

**Eight Gaussian** energy function (Dai et al., 2020) consists of eight equally spaced Gaussian components arranged in a circular pattern:

$$U(x, y) = \sum_{i=1}^{8} \frac{- \left[ (x - x_i)^2 + (y - y_i)^2 \right]}{2\sigma^2},$$

where the centers $(x_i, y_i)$ are located at $(\pm 1, 0)$, $(0, \pm 1)$, and $\left( \pm \frac{\sqrt{2}}{2}, \pm \frac{\sqrt{2}}{2} \right)$, with $\sigma = 1.0$ for all modes. This function presents multiple well-separated basins of attraction, testing the algorithm's ability to explore disconnected modes effectively.

**Moon** energy function creates a complex, asymmetric landscape resembling a crescent shape:

$$U(x, y) = -\frac{1}{10} y^4 - \frac{1}{2} \left( 4x - y^2 + \frac{24}{5} \right)^2.$$

With deep valley and steep ridge features, this non-convex structure evaluates the algorithm's capacity to explore non-uniform, curved regions and traverse narrow channels between high-energy barriers.

**Two Moons** energy function describes a landscape with two prominent crescent-shaped modes separated by a low-energy region:

$$U(x, y) = -\frac{2}{25} \left( x^2 + y^2 - 2 \right)^2 + \log \left[ e^{-\frac{1}{2} \left( \frac{5x-4}{4} \right)^2} + e^{-\frac{1}{2} \left( \frac{5x+4}{4} \right)^2} \right]$$

This function challenges MCMC methods to jump between distinct modes, and tests their efficiency in exploring multi-modal distributions where modes are not directly connected.

**Twist** energy function represents a twisted sinusoidal landscape where energy levels change smoothly along a sinusoidal curve:

$$U(x, y) = -\frac{1}{2} \left[ y - \sin \left( \frac{\pi x}{2} \right) \right]^2.$$

The narrow, twisted valleys require careful gradient following, which tests the algorithm's ability to sample from highly structured, nonlinear regions of the energy landscape.

**Flower** energy function combines radial symmetry with angular variations, forming a complex landscape with a petal-like structure:

$$U(x, y) = \sin \left( \sqrt{x^2 + y^2} \right) + \cos \left( 5 \tan \left( \frac{y}{x} \right) \right).$$

With multiple local minima and ridges radiating from the center, this intricate landscape tests the algorithm's capability to explore multi-modal, rotationally symmetric energy landscapes with sharp transitions between regions.

In our experiments, we evaluate different samplers using Kullback-Leibler (KL) divergence and Maximum Mean Discrepancy (MMD).

**KL divergence** measures the difference between two probability distributions. Given two distributions $\pi$ and $\tilde{\pi}$, the KL divergence is defined as:

$$D_{KL}(\pi \parallel \tilde{\pi}) = \sum_{\theta \in \Theta} \pi(\theta) \log \frac{\pi(\theta)}{\tilde{\pi}(\theta)},$$

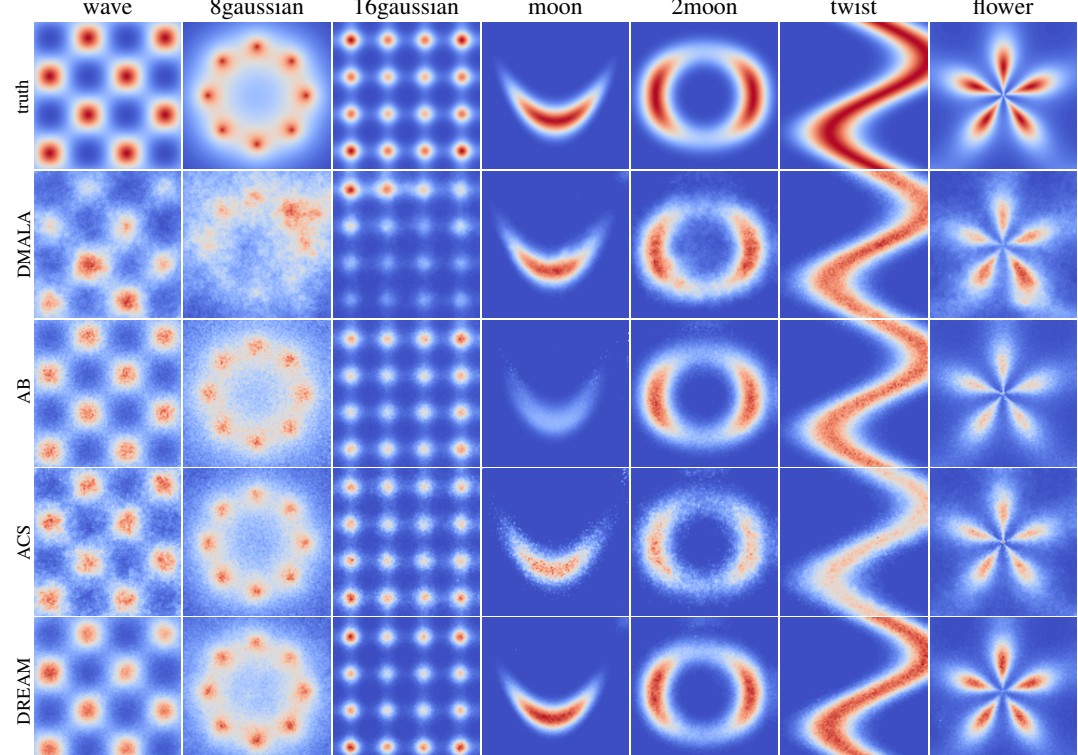

Figure 5: Visualization of the true energy function and the empirical energy function yield by DMALA, AB, ACS, and DREAM. Seven energy functions are tested here, which include wave, eight Gaussians, sixteen Gaussians, moon, two moons, twist, and flower energy functions. Red colors denote high-density regions, and blue colors represent low-density regions.

where $\pi(\theta)$ represents the probability of $\theta$ under the target distribution, and $\tilde{\pi}(\theta)$ represents the probability of $\theta$ under the empirical distribution from the samplers. This metric quantifies how much information is lost when $\tilde{\pi}$ is used to approximate $\pi$, with lower values indicating better performance.

**MMD** is a kernel-based test used to compare distributions. It is computed as:

$$\text{MMD}^2(\pi, \tilde{\pi}) = \mathbb{E}_{x,x'\sim\pi}[k(x, x')] + \mathbb{E}_{y,y'\sim\tilde{\pi}}[k(y, y')] - 2\mathbb{E}_{x\sim\pi,y\sim\tilde{\pi}}[k(x, y)],$$

where $k(x, y)$ is a positive-definite kernel function. MMD measures the similarity between the empirical distributions of the generated and target samples.

In practice, however, directly computing MMD is computationally expensive. Therefore, we use an approximation based on Random Fourier Features (RFF) (Rahimi & Recht, 2007).

For two distributions $\pi$ and $\tilde{\pi}$, we first map the data samples $X \sim \pi$ and $Y \sim \tilde{\pi}$ to a new feature space using the random Fourier transformation: $\phi(X) = \sqrt{\frac{2}{D}} \cos(WX^T + \tilde{b})$, where $W \in \mathbb{R}^{D\times\mathbf{d}}$ are random Gaussian variables sampled from $\mathcal{N}\left(0, 1/\tilde{\sigma}^2\right)$, and $\tilde{b}$ are random uniform variables in the range $[0, 2\pi]$. The parameter $\tilde{\sigma}$ controls the kernel bandwidth, and $D$ is the number of random features. Once mapped, the empirical mean feature embeddings for $X$ and $Y$ are computed for both distributions $\mu_X = \frac{1}{n} \sum_{i=1}^{n} \phi(X_i)$, $\mu_Y = \frac{1}{m} \sum_{i=1}^{m} \phi(Y_i)$. Finally, the MMD is approximated by the squared difference of the mean embeddings:

$$\text{MMD}^2(\pi, \tilde{\pi}) \approx \|\mu_X - \mu_Y\|^2.$$

This approach allows us to efficiently compute the MMD between two distributions using RFFs.

To evaluate the effectiveness of the proposed sampler, we explore its performance on a set of non-convex discrete energy landscapes that vary in complexity and multimodality. We further compare it with baselines such as DMALA, ACS, and AB. Unless specified otherwise, the default temperature for each sampler is set at 1.0. DMALA is implemented with a step size of 0.15. AB is used with parameters $\sigma = 0.10$ and $\alpha = 0.50$. For ACS, a cyclical step size scheduler with an initial step size

of 0.60 across 10 cycles is applied. DREAM uses small and large step sizes of 0.15 and 0.60 and temperatures of 1.0 and 5.0.

Figure 5 illustrates a comparison between the empirical distributions obtained from different samplers and the ground truth (top row). A clear distinction can be observed between the discrete samplers in terms of their ability to capture the full complexity of the landscape. From the figure, DREAM produces the most balanced and comprehensive empirical distribution, which captures all significant modes of the energy landscape. The improvements are significant in tasks of approximating wave and multi-Gaussian energy functions. While other discrete samplers (such as DMALA, AB, ACS) fail to capture all modes, DREAM exhibits the most comprehensive exploration, as reflected in the uniformity of the empirical distribution across all modes. Its ability to distribute samples effectively, even in the presence of disconnected modes and sharp energy barriers, demonstrates its robustness in navigating complex discrete energy landscapes. By contrast, DMALA shows a heavy concentration of samples around certain modes, which indicates that it struggles to escape local minima. This leads to poor coverage of the landscape and a lack of diversity in the sampled regions. ACS and AB perform better in terms of covering multiple modes but still show uneven sample distributions. Some modes are under-sampled, while others are over-sampled, particularly in regions with shallow energy gradients.

### E.2 Sampling from Ising Models

We sampled from the Ising model using multiple samplers with a default temperature of 1.0, unless otherwise specified. DULA utilized a step size of 0.20, while DMALA had a step size of 0.40. For ACS, we employed a cyclical step size scheduler with 10 cycles and an initial step size of 0.30. ACS with MH corrections used an initial step size of 5.0. For DREXEL, small and large step sizes were set at 0.15 and 0.50, with temperatures at 1.0 and 5.0. DREAM followed a similar temperature schedule, with a small step size of 0.35 and a large size of 0.50.

In this study, different samplers are evaluated with log Root Mean Square Error (log RMSE). Log RMSE evaluates prediction accuracy for comparing the true values from $\pi$ and the predictions from $\tilde{\pi}$, which can be adapted as:

$$\log \text{RMSE} = \log\left(\sqrt{\frac{1}{n}\sum_{i=1}^{n}(\pi(\boldsymbol{x}_i) - \tilde{\pi}(\boldsymbol{x}_i))^2}\right),$$

where $\pi(\boldsymbol{x}_i)$ is the true value under the target distribution, and $\tilde{\pi}(\boldsymbol{x}_i)$ is the corresponding approximation from the empirical distribution.

We further examine the influence of bias correction terms in DREXEL and DREAM with bias corrections (bDREXEL and bDREAM) on the experimental results of Ising models. For comparison, we also evaluate DREXEL and DREAM, which incorporate historical energy corrections. Specifically, we modify the swap function (10) by directly adding bias corrections and adjusting $\sigma^2$ in (9). Each experiment is repeated 20 times with different random seeds, and the average and standard deviation of log RMSE are reported. The results are shown in Figure 6.

The results demonstrate that DREAM achieves the lowest log RMSE values when corrections are small, which indicates the minimal influence of bias correction on its performance. In contrast, the performance of bDREXEL and bDREAM becomes better with increasing $\sigma^2$, which suggests its performance may heavily rely on a good selection of bias correction. bDREAM and DREAM show comparable log RMSEs, with log RMSE across different bias correction magnitudes. However, DREAM yields the lowest log RMSE with no correction, while bDREAM has the lowest one when the correction increases.

### E.3 Sampling from Restricted Boltzmann Machines

We trained RBMs with the Adam optimizer with a learning rate of 0.001, over 1,000 iterations, and a batch size of 128. For training, we used contrastive divergence (CD), which approximates the log-likelihood gradient by performing $k = 10$ Gibbs sampling steps. The gradient of the log-likelihood for an RBM is given by:

$$\nabla_{\boldsymbol{\theta}} \log P_{\boldsymbol{\theta}}(\boldsymbol{x}) = \mathbb{E}_{P_{\text{data}}}[\nabla_{\boldsymbol{\theta}} \log P_{\boldsymbol{\theta}}(\boldsymbol{x})] - \mathbb{E}_{P_{\boldsymbol{\theta}}}[\nabla_{\boldsymbol{\theta}} \log P_{\boldsymbol{\theta}}(\boldsymbol{x})],$$

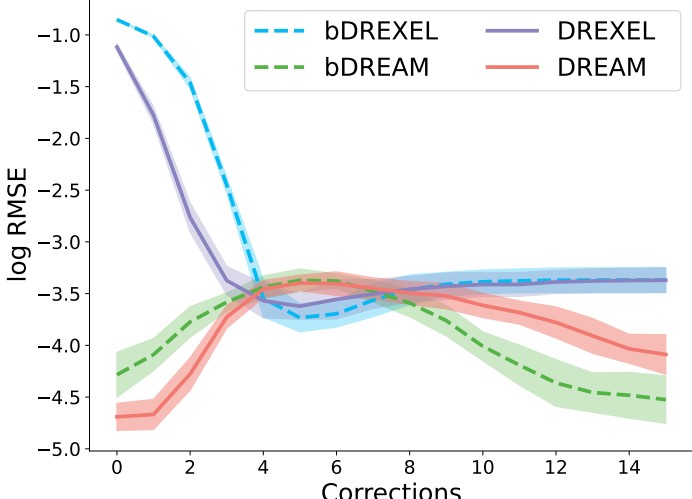

Figure 6: Log RMSE results for bDREXEL, bDREAM, DREXEL, and DREAM under varying correction terms. Solid and dashed lines denote the log RMSE average values, and a lighter shade represents 95% confidence intervals. Lower log RMSE values indicate better performance.

where $\boldsymbol{x}$ is the visible layer, and $\boldsymbol{\theta}$ are the model parameters. The first term corresponds to the data distribution, while the second term is the expectation under the model's distribution. It should be noted that direct computation of the model distribution expectation is expensive, and CD approximates the second term by running $k$-step Gibbs sampling to obtain samples from the model. This approximation enables efficient training of RBMs by focusing on the contrast between the observed and modeled distributions.

To sample from the RBMs, we employed several discrete samplers at a default temperature of 1.0 unless otherwise noted. DMALA used a step size of 0.15, and ACS applied a cyclical step size scheduler with 10 cycles, starting at a step size of 0.50. The Any-scale Balanced (AB) sampler was configured with $\sigma = 0.10$ and $\alpha = 0.50$. For DREAM, the small and large step sizes were set between 0.15–0.20 and 0.40–0.50, with temperatures set at 1.0 and 2.0.

E.4    LEARNING DEEP ENERGY-BASED MODELS

We trained Deep EBMs using a ResNet-64 backbone and optimized the model with the Adam optimizer at a fixed learning rate of 0.001 without gradient clipping. The model was trained for 50,000 iterations with a batch size of 256. we employed Persistent Contrastive Divergence (PCD) to approximate the intractable likelihood gradient, which builds upon standard contrastive divergence by maintaining persistent Markov chains throughout training. It allows for more stable and accurate sampling. Specifically, the model's log-likelihood gradient is given by:

$$\nabla_{\boldsymbol{\theta}} \log P_{\boldsymbol{\theta}}(\boldsymbol{x}) = \mathbb{E}_{P_{\text{data}}}[\nabla_{\boldsymbol{\theta}} \log P_{\boldsymbol{\theta}}(\boldsymbol{x})] - \mathbb{E}_{P_{\boldsymbol{\theta}}}[\nabla_{\boldsymbol{\theta}} \log P_{\boldsymbol{\theta}}(\boldsymbol{x})],$$

where the second term (the model expectation) is intractable. PCD approximates this by updating samples across training iterations via Gibbs sampling, which ensures that the Markov chain does not restart after each parameter update. Additionally, a replay buffer containing 1,000 past samples is used to further stabilize training. The buffer stores past model samples and reuses them to reduce variance, thus improving both the efficiency and stability of the learning process.

To evaluate Deep EBMs, we applied Annealed Importance Sampling (AIS) with DULA to estimate the test log-likelihoods. AIS is a technique used to estimate partition functions by smoothly interpolating between a known distribution and the target distribution. This is achieved by introducing a sequence of intermediate distributions:

$$P_t(\boldsymbol{x}) = \frac{1}{Z_t} \exp(-\beta_t E(\boldsymbol{x})),$$

where $t$ denotes the current annealing step, $E(\boldsymbol{x})$ is the energy function, $Z_t$ is the partition function, and $\beta_t$ is a temperature that gradually transitions between 0 and 1 over the course of the annealing

process. When $\beta_t = 0$, the intermediate distribution is identical to the proposal distribution (which we can sample from easily). When $\beta_t = 1$, the intermediate distribution becomes the target distribution, which is more complex and generally intractable to sample directly. To adjust $\beta_t$, we typically choose a monotonic schedule that increases smoothly from 0 to 1 over the course of the AIS process. A common choice is a linear interpolation ($\beta_t = t/T$, $t = 0, 1, 2, \ldots, T$) or an exponential schedule ($\beta_t = (t/T)^2$), where $\beta_t$ increases evenly across $T$ annealing steps. AIS computes an estimate of the partition function by sampling from these intermediate distributions and adjusting the importance weights over time:

$$\tilde{Z}_\theta = Z_0 \prod_{t=1}^{T} \frac{P_t(\boldsymbol{x})}{P_{t-1}(\boldsymbol{x})},$$

where $Z_0$ is an initial distribution that is easy to sample from, which serves as a starting point for the annealing process. Typically, $Z_0$ is chosen to be the partition function of a simple proposal distribution $p_0(x)$, which is often a uniform distribution or a Gaussian distribution with parameters that are easy to compute. In our experiments, we used AIS with 40 samples and 30,000 annealing steps. DULA was configured with a step size of 0.08 and a temperature of 1.00. Detailed hyperparameters for training Deep EBMs are listed in Table 3.

| Static MNIST | DULA | DMALA | bDREXEL | bDREAM | DREXEL | DREAM |
|---|---|---|---|---|---|---|
| Step size | 0.08 | 0.10 | 0.05 | 0.05 | 0.11 | 0.10 |
| | - | - | 0.15 | 0.15 | 0.25 | 0.30 |
| Temperature | 1.0 | 1.0 | 1.0 | 1.0 | 1.0 | 1.0 |
| | - | - | 5.0 | 5.0 | 5.0 | 5.0 |
| Correction | - | - | 0.00 | 0.00 | 0.00 | 0.00 |

| Dynamic MNIST | DULA | DMALA | bDREXEL | bDREAM | DREXEL | DREAM |
|---|---|---|---|---|---|---|
| Step size | 0.08 | 0.10 | 0.05 | 0.05 | 0.11 | 0.11 |
| | - | - | 0.15 | 0.15 | 0.25 | 0.25 |
| Temperature | 1.0 | 1.0 | 1.0 | 1.0 | 1.0 | 1.0 |
| | - | - | 5.0 | 5.0 | 5.0 | 5.0 |
| Correction | - | - | 0.00 | 0.00 | 0.00 | 0.00 |

| Omniglot | DULA | DMALA | bDREXEL | bDREAM | DREXEL | DREAM |
|---|---|---|---|---|---|---|
| Step size | 0.08 | 0.10 | 0.05 | 0.05 | 0.08 | 0.08 |
| | - | - | 0.15 | 0.15 | 0.15 | 0.15 |
| Temperature | 1.0 | 1.0 | 1.0 | 1.0 | 1.0 | 1.0 |
| | - | - | 5.0 | 5.0 | 5.0 | 5.0 |
| Correction | - | - | 0.00 | 0.00 | 1.00 | 0.00 |

| Caltech | DULA | DMALA | bDREXEL | bDREAM | DREXEL | DREAM |
|---|---|---|---|---|---|---|
| Step size | 0.08 | 0.10 | 0.05 | 0.05 | 0.08 | 0.08 |
| | - | - | 0.15 | 0.15 | 0.20 | 0.20 |
| Temperature | 1.0 | 1.0 | 1.0 | 1.0 | 1.0 | 1.0 |
| | - | - | 5.0 | 5.0 | 5.0 | 5.0 |
| Correction | - | - | 0.00 | 0.00 | 0.00 | 0.00 |

Table 3: Hyper-parameters used in learning Deep EBMs. From top to bottom, hyper-parameters in Static MNIST, Dynamic MNIST, Omniglot, and Caltech Silhouettes are recorded.

For a consistent comparison with previous works (Zhang et al., 2022b), we record its log-likelihood on the test set after 50,000 iterations. In general, Table 4 yields consistently lower log-likelihood than the results in Table 2 since they train with more iterations. But similar trends are shown in

Table 2 as well: DREAM consistently achieved the highest log-likelihoods across all datasets, with significant improvements on Omniglot and Caltech Silhouettes. This demonstrates the proposed swap mechanism in (10) effectively corrects imbalance, which leads to improved log-likelihood estimates across diverse image datasets. For the MNIST datasets, both DREXEL and DREAM showed competitive performance. bDREXEL and bDREAM generally perform worse than DREAM, with consistently lower log-likelihoods, which further confirms the advantage of historical energy corrections. These findings suggest that incorporating MH steps is crucial for enhancing performance in discrete sampling tasks. DULA and DMALA exhibit the lowest performance overall, which emphasizes the benefits of MH steps and the need to consider DREXEL and DREAM to enhance exploration.

| Dataset | DULA | DMALA | bDREXEL | bDREAM | DREXEL | DREAM |
| --- | --- | --- | --- | --- | --- | --- |
| Static MNIST | -79.672 | -77.581 | -77.212 | -76.840 | -75.685 | -74.883 |
| Dynamic MNIST | -81.144 | -79.411 | -81.273 | -81.043 | -71.091 | -70.905 |
| Omniglot | -114.203 | -109.095 | -94.382 | -90.807 | -89.971 | -89.643 |
| Caltech Silhouettes | -102.546 | -98.554 | -96.073 | -93.969 | -89.764 | -86.624 |

Table 4: EBM learning results (log-likelihood) on the test set after 50,000 iterations.

Here we provide the generated results (Figure 7) from DREAM across Static MNIST, Dynamic MNIST, Omniglot, and Caltech Silhouettes. These images demonstrate the ability of trained deep EBMs to capture the underlying data distribution. The deep EBM is capable of producing high-quality samples that visually resemble the training data, which indicates that the learned energy function effectively models the complex, high-dimensional structure of the data.

Figure 7: Deep RBMs sampling results from DREAM. **Top Left:** Static MNIST; **Top Right:** Dynamic MNIST; **Bottom Left:** Omniglot; **Bottom Right:** Caltech Silhouettes.

