# OpenReview forum: "Exploring Non-Convex Discrete Energy Landscapes: A Langevin-Like Sampler with Replica Exchange"
_ICLR.cc/2025/Conference — ICLR 2025 Conference Withdrawn Submission_

### Official Review · Reviewer_XqY2 · 2024-11-03

**Soundness:** 3
**Presentation:** 4
**Contribution:** 3
**Rating:** 5
**Confidence:** 4

**Summary:**

In this paper, the authors tackle the task of sampling from high-dimensional discrete distributions with multimodality characteristics (i.e. non log-concave distributions) whose energy is assumed to be differentiable. This is a computational challenge since it requires to be able to locally explore the target distribution and alternating between the modes without introducing any asymptotic bias. Inspired by the rich literature in continuous state spaces, the authors propose a novel MCMC sampler, DREXEL, which can be seen as a discrete version of the well known Replica Exchange Langevin algorithm. In essence, DREXEL builds 2 Markov chains, each one targeting a tempered version of the target distribution: one with temperature 1 (i.e., the target distribution) and the other with high temperature (i.e., smoother version of the target). Each of these Markov chains consists in discrete Langevin steps, independently made on all coordinates (in order to explore the local landscape), which may be combined to Metropolis-Hastings (MH) corrections (in this case DREXEL is referred to as DREAM). To enable global exploration, swaps between the current states of these MCs may occur, based on a tractable MH acceptance rate. To validate their approach, the authors provide convergence results in the case where the target distribution is log-quadratic, which demonstrate zero asymptotic bias and better convergence rate than standard local sampler. Then they apply their method to a variety of discrete sampling settings with multimodality including Ising models and EBMs on binarized images.

**Strengths:**

- The current paper is well written and easy to follow; the authors pay attention to recall in their introduction the main elements previously obtained in the literature before explaining how they can be adapted to their setting. Overall, they bring intuition on each methodological brick and each theoretical statement, which makes the reading pleasant.
- This is a complete methodological work, detailing both theoretical and numerical results for a challenging sampling task (i.e. including multimodality), and therefore, it may be highly valuable for the sampling community.
- A large variety of numerical settings is considered in the experiments with convincing results, although they suffer from main weaknesses (see next section)

**Weaknesses:**

In my opinion, the main weaknesses come from the numerical evaluation of the method:
- The setting of the three main hyperparameters of DREXEL/DREAM (the local Langevin step-size, the highest temperature and the swap intensity) is not discussed anywhere in the manuscript, whereas it is critical for real world use. In particular, the authors do not detail how they tuned their hyperparameters for their experiments: in principle, one should not know in advance the ground truth samples for tuning. If this is the case, this should be well highlighted.
- No ablation study of the hyperparameters is given anywhere, which makes hard to tell how much they are critical.
- The initialization of the algorithm is not given in each numerical experiment, while it is known to have a high influence on MCMC results.
- The current numerical experiments do not rely on sampling metrics that could assess how well the true relative weights of the target modes are estimated (for distributions with isolated modes such as 8 gaussians or wave) beyond recovering the location of the modes, which is the main challenge for multimodal sampling. In particular, integrated probability metrics (such as MMD, KL, used in the first experiment) may not reflect mode collapse, as explained in [1].
- The numerical results not contain error quantifications computed on several experiment runs, which hurts their statistical significance. In particular, numerical results from various methods are really close to each other in each setting and may be considered similar by repeating runs.

[1] Beyond ELBOs: A Large-Scale Evaluation of Variational Methods for Sampling. Blessing et al. 2024.

**Questions:**

- Have you considered increasing the number of Langevin steps before applying the swap ? In the current version, there is only one Langevin step for one RE step which definitely helps the mixing ; however, having a small number of Langevin steps may actually hurt the local exploration.
- Have you considered adaptive Langevin step-size based on the result of the MH step for DREAM ? This is widely used for continuous state space and may be valuable in the current setting, as proposals are made independently on each coordinate.
- Why don't you consider more than 2 temperatures in the RE setting as it is commonly made for continuous state spaces, which may speed up the mixing of the Markov chains ?
- Why don't you consider a sample-based metric for the EBM experiment as made for the Boltzmann machine ? In particular, I think that estimating the partition function with MCMC to obtain an estimate of the EBM log likelihood may bring high statistical uncertainty to the results.
- How did you initialize the sampling algorithms in each numerical setting ?
- Since you used an annealed version of DULA (ie, discrete version of annealed Langevin dynamics) to estimate the partition function of EBMs, why don't you use this sampling algorithm as a competing method ?
- How many iterations of DREAM did you use to sample from one MNIST image once the EBM is trained?

**Suggestions**
- I think that the assumptions on the target distribution should be more highlighted in the text.

---

> ### Author Response · Authors · 2024-11-24
> **Response to reviewer XqY2 (1/2)**
>
> We appreciate your time and great efforts in reviewing. Please find our responses to the questions below.
>
> > **The setting of the three main hyperparameters of DREXEL/DREAM (the local Langevin step size, the highest temperature, and the swap intensity) is not discussed anywhere**
>
> We outline below the practical guidelines we followed for selecting these key hyperparameters:
> - Low-Temperature Sampler:
>   - We set $\tau_1 = 1.0$ by default to approximate the unaltered target distribution.
>   - For DREAM, $\alpha_1$ was selected to achieve an acceptance rate of approximately 70%–80% in the MH steps to balance stability and sampling efficiency.
>   - For DREXEL, $\alpha_1$ was chosen to be smaller than that used in DREAM to ensure stability and guarantee convergence in the absence of MH corrections.
>
> - High-Temperature Sampler:
>   - We gradually increase $\tau_2$ while monitoring the sampler’s ability to escape local minima and explore effectively.
>   - Larger step sizes $\alpha_2$ are paired with $\tau_2$ to enhance exploratory behavior, with adjustments based on performance in navigating complex energy landscapes.
>
> - Swap Intensity ($\rho$) is tuned to achieve a swap rate between 5%–20%.
>
> > **No ablation study of the hyperparameters is given anywhere**
>
> We acknowledge that our current work includes only a basic ablation study, which is presented in Appendix E.2. We will expand these studies in our future revisions to provide a more detailed analysis of the impact of hyperparameters.
>
> > **The initialization of the algorithm is not given in each numerical experiment**
>
> We summarize the initialization methods employed in our experiments as follows:
> - Synthetic 2D Tasks: The starting sample is initialized by uniformly and randomly selecting a position within the 2D grid.
> - Ising Models: Initial spin configurations are drawn from a Bernoulli distribution, where the logits are set to $P(\theta_i = +1) \approx 0.6$ and $P(\theta_i = -1) \approx 0.4$.
> - RBMs: The sample distribution is initialized using a Bernoulli distribution with equal probabilities for all visible units.
> - Deep EBMs: Initial samples are drawn from a Bernoulli distribution parameterized by the empirical mean of the first batch in the dataset.
>
> > **Why not consider sampling metrics to reflect mode collapse in the 2d synthetic tasks?**
>
> We appreciate the reviewer’s insightful suggestion and the reference to relevant work. Integrated probability metrics indeed emphasize a subset of modes, which potentially leads to an underestimation of mode collapse. We will incorporate metrics in our revision that more effectively detect mode collapse and assess the ability of the sampling methods to preserve the true relative weights of the target modes.
>
> > **The numerical results do not contain error quantifications**
>
> We acknowledge that error quantifications were not included for all experiments due to time constraints. We will address this in our revision by reporting error analyses for the results to provide a more robust evaluation.
>
> > **Have you considered increasing the number of Langevin steps before applying the swap?**
>
> We follow the traditional reMCMC framework and typically set the swap rate between 5%-20%, which corresponds to approximately 5-20 Langevin steps between successive swaps.

---

> > ### Author Response · Authors · 2024-11-24
> > **Response to reviewer XqY2 (2/2)**
> >
> > > **Have you considered adaptive Langevin step size based on the result of the MH step for DREAM? This may be valuable in the current setting, as proposals are made independently on each coordinate.**
> >
> > One challenge to implementing adaptive Langevin step sizes is maintaining consistency for each coordinate with the MH step. Specifically, the MH step requires a proposal that uses a uniform step size across all coordinates, which complicates the integration of coordinate-wise adaptive step sizes.
> >
> > When implementing DREAM, we adjust the step size based on the suggested optimal MH acceptance rate from [3], which already demonstrated good performance in our experiments. Note that our method is fully compatible with adaptive step sizes, and we recognize the potential value of exploring this direction in future work.
> >
> > > **Consider more than 2 temperatures in the replica exchange Langevin algorithms**
> >
> > Empirical evidence from prior works in continuous settings [1, 2] indicates that employing multiple temperature levels can enhance performance, particularly when sampling from highly multi-modal distributions. This strategy is indeed promising in discrete settings, where multi-scale exploration could address challenges posed by highly non-convex landscapes.
> > However, certain techniques effective in continuous domains may not directly translate to discrete scenarios. For instance, window-wise corrections [1] used in swap mechanisms may require significant adaptation or alternative formulations for discrete sampling tasks. Swap schemes such as stochastic or deterministic even-odd swaps may have unclear efficacy in discrete domains and warrant further investigation.
> >
> > We recognize that extending multiscale methods to discrete settings involves non-trivial challenges. Exploring these adaptations and evaluating their benefits will be a promising direction for our future work.
> >
> > > **Why not consider a sample-based metric for the EBM experiment? Estimating the partition function with MCMC to obtain an estimate of the EBM log-likelihood may bring high statistical uncertainty to the results.**
> >
> > While sample-based metrics are commonly used in model evaluation, they often depend on the sample quality. This can introduce biases when the sampling methods fail to converge adequately in high-dimensional spaces or under constrained computational budgets.
> >
> > To mitigate the high statistical uncertainty associated with estimating the EBM log-likelihood via AIS, we ensure a sufficient number of intermediate distributions, which allows for a smoother transition between the base and target distributions. Also, a large number of MCMC steps are employed between these intermediate distributions to improve convergence and reduce variance in the estimation process. We believe that these precautions provide a robust framework for reducing both variance and bias when estimating the EBM log-likelihood.
> >
> > > **How many iterations of DREAM did you use to sample from one MNIST image once the EBM is trained?**
> >
> > We run 300,000 iterations of AIS, where each AIS step utilizes DULA to perform 500 sampling iterations.
> >
> > > **Consider using an annealed version of DULA as a competing method**
> >
> > In our experiments, AIS is employed during log-likelihood estimation to approximate the normalizing constant $Z$ in Eq. (1). However, this is separate from the training process, which relies on gradient-based sampling and does not require approximating $Z$.
> >
> > If the suggestion refers to annealing the step sizes of DULA, we note that smaller step sizes can make the sampler easily trapped in local regions when sampling from discrete distributions. To address this, we adopt a fixed step size in our experiments to facilitate escape from local traps and ensure better exploration of the energy landscape.
> >
> > > **Highlight the assumptions on the target distribution**
> >
> > We appreciate the reviewer’s suggestion and highlighted the assumptions in our revision.
> >
> > **References**
> >
> > [1] Non-reversible Parallel Tempering for Deep Posterior Approximation. AAAI 2023
> >
> > [2] Non-reversible Parallel Tempering: A Scalable Highly Parallel MCMC Scheme. Journal of the Royal Statistical Society: Series B 2022.
> >
> > [3] MCMC Using Hamiltonian Dynamics. arXiv:1206.1901

---

> ### Comment · Reviewer_XqY2 · 2024-11-24
> **Answer to the rebuttal**
>
> Thank you for your answer. Regarding your response, I may have additional questions:
>
> > We gradually increase $\tau_2$ while monitoring the sampler’s ability to escape local minima and explore effectively.
>
> In practice, how do you evaluate this 'ability' of exploration without having access to ground truth samples ? Which metric do you use ?
>
> > Swap Intensity is tuned to achieve a swap rate between 5%–20%
>
> Could you explain the origin of this setting for the swap rate ?
>
> About the use of several temperatures in the RE mechanism: the authors seem to claim that using more than two temperatures may need non-trivial adaptation for their algorithm. However, in the case of deterministic even/odd swaps, the swap acceptance (10) would write the same between considered states, and the sampling part would still occur in parallel: where would the difficulty come from ?
>
> > While sample-based metrics are commonly used in model evaluation, they often depend on the sample quality. This can introduce biases when the sampling methods fail to converge adequately in high-dimensional spaces or under constrained computational budgets.
>
> I am quite confused about this answer: I don't understand the computational difference between estimating the partition function which requires AIS and directly sampling from the EBM, which seems to require AIS as you suggest below. You seem to suggest that sampling from the EBM may fail; in that case, this would directly question the accuracy of the partition function estimation ?
>
> > We run 300,000 iterations of AIS, where each AIS step utilizes DULA to perform 500 sampling iterations.
>
> So you don't use DREAM to sample from the EBM in order to obtain samples after training ? Then, why using DREAM to sample from the EBM when training ?
>
> About the question of using AIS as competing method: sorry if I was not clear in my first question, I'd like to clarify it. I was suggesting the authors to use AIS as a multi-level sampling scheme (ie Sequential Monte Carlo) to compare with RE (since they are often opposed in the continuous setting: sequential vs parallel) : the sampling scheme would proceed exactly the same as done for the EBM evaluation (sequentially, from highest temperature to lowest temperature with DULA steps, while keeping the AIS importance weights to readjust particles). Have you tried to implement it as competitor for RE ? It seems that you have used this method to obtain samples from the EBM at least (see above).
>
>
> Finally, I'll be happy to increase my score upon the release of a revision that main contain:
> - the ablation study on the hyperparameters
> - the use of mode collapse / mode weight metrics
> - uncertainty errors on numerics (which would be notably meaningful for the EBM experiment)

---

### Official Review · Reviewer_Vjro · 2024-11-03

**Soundness:** 2
**Presentation:** 3
**Contribution:** 2
**Rating:** 3
**Confidence:** 3

**Summary:**

This paper adapts the replica exchange technique from continuous gradient-based samplers to discrete versions, which expects to have a better gradient complexity to achieve convergence for dissipative target distributions. From an intuition perspective, this paper hopes to utilize different temperatures (or step sizes of each iteration) and exchange them for escaping the trap's local modes of non-convex discrete energy landscapes.

**Strengths:**

1. The problem about drawing samples from a discrete energy landscape is very interesting and worth studying
2. The core technical contribution of this paper may be the design of the swap function shown in Eq.(10), which is original and seems to have a good performance in various experiments.
3. This paper is well written; the intuition, implementation, and experiments are clear, which makes this paper easy to follow.

**Weaknesses:**

I have some serious concerns with respect to the theoretical part of this paper.
1. Theorem 2 considers the target distribution, which is affected by both $\theta_1$ and $\theta_2$ corresponding to the low or high-temperature particles. Actually, in such a discrete sampling setting, the return of Alg.1 is only $\theta_1$ corresponding to the low-temperature particles, which does not match the theory shown in Theorem 2.
2. Essentially, authors should provide the TV distance convergence between the marginal distribution of $\theta^{(1)}_i$ (denoted as $\tilde{\pi}\_n^{(1)}$) and $\theta\_{*}\sim \pi\propto(U(\theta))$. However, except for $\tau_1=1$, the target distribution provided in Theorem 2 cannot be considered as an approximation of $\pi$ even only considering its marginal distribution, i.e., $\exp(U(\theta^{(1)})/\tau_1)$ of $\theta^{(1)}$. Since additional factor $\tau_1$ exists .
3. According to the chain rule of TV distance, the TV distance provided in Theorem. 2 is only an upper bound of the TV distance between the output distribution of $\theta^{(1)}$ and $\pi$. This means the authors should claim the advantages of convergence improvement come from the sampling algorithm rather than the TV gap between joint distribution and marginal distribution, i.e., $TV(\pi^\prime, \tilde{\pi}^\prime_n) - TV(\pi, \tilde{\pi}^{(1)}_n)$.
4. Moreover, the proof of this paper has a severe problem in Line 1013—Line 1016. We can easily find that $0<\rho<1$, and the last line of Eq.(18) can hardly be upper bounded by arbitrary small $\epsilon$.  Because the last term is $(\mathcal{B}+B)\cdot 1/(1-\rho)$ where both $\mathcal{B}$ and $B$ can be founded in Assumption. It means they can be considered as a constant. Besides, the inequality $1/(1-\rho)>1$ is established. More explicitly, it can be easily found that the last inequality in Eq.(19) requires $\epsilon>(\mathcal{B}+B)$, which is unacceptable in convergence analysis.
5. The convergence improvement in the analysis is extremely limited since there is a log dependence with respect to such an improvement, which may hardly be distinguished in practice.

**Questions:**

1. In the synthetic data, it is relatively easy to determine the different temperatures. However, determining the scaling of different temperatures is difficult. Could authors give some suggestions for tuning the hyperparameters?
2. From an intuitive perspective, a Langevin dynamic with a decreasing step size seems to achieve a similar performance (get rid of the local traps) under a large initial step size. Could authors have a discussion or additional experiments to compare with those methods?

I am willing to raise my score if the authors solve my concerns.

---

> ### Author Response · Authors · 2024-11-24
> **Response to reviewer Vjro**
>
> We appreciate your constructive feedback. Please find our responses below.
>
> > **The return of Alg. 1 is only $\theta_1$, but Theorem 2 analyze the target distribution affected by $\theta_1$ and $\theta_2$.**
>
> We agree that when comparing to DULA, our analysis requires an additional step to explicitly show the convergence of the marginal distribution.
>
> The convergence of the marginal distribution of $\theta_1$ can be trivially bounded by the convergence of the joint distribution of $(\theta_1, \theta_2)$ using the triangle inequality, as the total variation distance between the marginals is directly bounded by that of the joint distributions. We will revise our analysis to clarify this point in our revisions.
>
> > **Theorem 2 cannot be considered as an approximation of $\pi$ considering its marginal distribution due to the additional factor $\tau_1$.**
>
> We want to clarify that $\tau_1$ is typically set to 1.0, which ensures that the low-temperature sampler approximates the unaltered target distribution. When $\tau_1 \neq 1.0$, the sampler produces a biased distribution that deviates from the true target distribution. We appreciate the reviewer’s feedback and will clarify this point in the revision.
>
> > **The authors should claim the advantages of convergence improvement come from the sampling algorithm rather than the TV gap between joint distribution and marginal distribution**
>
> Thank you for the feedback. We agree with this observation and will revise our analysis to emphasize that the convergence improvement arises from the sampling algorithm rather than the TV gap between the joint and marginal distributions.
>
> > **TV gap cannot be bounded by a small constant $\epsilon$ in Theorem 2.**
>
> We thank the reviewer for the feedback on our theoretical analysis. We acknowledge that Theorem 2 may be suboptimal. We will improve the theoretical part in future revisions.
>
> > **The convergence improvement in the analysis is extremely limited.**
>
> We acknowledge the reviewer’s concern regarding the improvement in analysis. We want to clarify that investigating the quantitative value of the spectral gap for replica exchange Langevin algorithms remains a fundamental challenge in this field. Recent work [1] demonstrated the potential for improving the spectral gap in continuous settings when employing replica exchange Langevin diffusion compared to vanilla Langevin diffusion. We anticipate that a similar acceleration effect would extend to discrete settings.
>
> Given the methodological focus of this paper, our primary goal is to bridge practical design principles with theoretical insights for discrete samplers. A comprehensive exploration of spectral gap quantification and its improvement in discrete settings is indeed an exciting direction for future research, which we plan to investigate further.
>
> > **Determining the scaling of different temperatures is difficult. Could authors give some suggestions for tuning the hyperparameters?**
>
> For low-temperature sampler:
> - We set $\tau_1 = 1.0$ to sample from the unaltered target distribution.
> - For DREAM, the step size $\alpha_1$ was chosen to guarantee around 70%-80% accept rate in MH steps.
> - For DREXEL, $\alpha_1$ should be smaller than that used for DREAM to maintain stability and ensure convergence.
>
> For high-temperature sampler:
> - Larger values of $\tau_2$ and $\alpha_2$ generally enhance the exploration capabilities.
> - We gradually increase $\tau_2$ and $\alpha_2$ while monitoring the sampler's ability to effectively explore the energy landscape.
>
> > **Could authors discuss and compare the method to consider decreasing step sizes in Langevin dynamics?**
>
> In continuous distributions, annealing step sizes in Langevin Monte Carlo is a well-established strategy to ensure convergence to equilibrium. However, this approach is less effective when sampling from discrete distributions due to the intrinsic discontinuities of the space. As the step size decreases, the sampler's ability to traverse between discontinuous states diminishes and it is easy to get trapped in local regions.
>
> To address this, the common approach adopts a larger, fixed step size to facilitate broader exploration and avoid local traps. This is particularly important for navigating highly non-convex discrete energy landscapes.
>
>
> **References**
>
> [1] Spectral gap of replica exchange langevin diffusion on mixture distributions. Stochastic Processes and their Applications 2022.

---

### Official Review · Reviewer_tAX1 · 2024-11-04

**Soundness:** 2
**Presentation:** 2
**Contribution:** 1
**Rating:** 6
**Confidence:** 4

**Summary:**

Two gradient-based discrete samplers (with and without MH corrections) are introduced by combining concepts from discrete Langevin samplers (DLS), which generalize Langevin MCMC to discrete space, and replica exchange Markov Chain Monte Carlo (reMCMC), which considers exchange between multiple replicas operating at different temperatures.

**Strengths:**

The numerical results are backed by tests on various model types, demonstrating statistically significant improvements over prior related methods.

**Weaknesses:**

While the approach is sound and results are well-presented, the work represents an incremental improvement, building heavily on previous methods, analyses, and benchmarks. It would be helpful to clarify a few points for better readability.

In terms of algorithmic progress, the new approach simply combines existing ideas (DLS and reMCMC). Moreover, the improvement in benchmark results is statistically significant but relatively modest. It is not entirely clear how much the setup (hyperparameter settings, etc.) of the benchmark affects the relative performance of the compared algorithms.

**Questions:**

1) What is the dependence of log RMSE on temperature τ in Ising model results (Fig. 3) ?
Are the conclusions of Fig. 3 the same for a lower target distribution temperature (e.g., τ=0.1) ?
What are the results for other spin connectivity exhibiting more frustration (e.g. fully connected Sherrington-Kirkpatrick problems) ?

2)  Although the new method seems effective, the rationale behind the selection of hyperparameters for each approach is not sufficiently justified. To ensure a fair comparison, additional details about the hyperparameter settings would be beneficial.

- How are temperature and step sizes chosen in each approach?
In particular, how is the gap between τ1 and τ2 chosen and how does it affect performance?

- Are DMALA and DULA comparison done at their respective optimal parameters (step size)?
How does log RMSE of Ising model results of DMALA and DULA change for step size from 0.1 to 0.5 ?

- Could the authors clarify the reasoning behind choosing, for E.g., a step size of 0.2 for DULA and 0.4 for DMALA on the Ising model?
Discussion the influence of hyperparameter selection would make the results more convincing.

3) The explanation of bias correction is not totally clear, specifically for methods like dDREXEL and bDREAM versus DREXEL and DREAM. While dDREXEL and bDREAM include a bias, do DREXEL and DREAM incorporate a bias correction as well (see Fig. 6)?

Others:

1) How is the bias 𝜎 σ in equation (9) chosen in practice?
2) What are the rejection rates for DREAM, and how do step size and temperature choices (𝜏1, 𝜏2) affect it?

Minor comments:

1) Missing standard errors in Tables 1 and 2
2) Typo: Asymptotic Convergence on Log-Quadratic “Distriubions"

---

> ### Author Response · Authors · 2024-11-24
> **Response to reviewer tAX1 (1/2)**
>
> We appreciate your thoughtful review. Please find our responses to the questions below.
>
> > **The new approach simply combines existing ideas (DLS and reMCMC)**
>
> While the approach builds upon DLS and reMCMC, this work addresses unique challenges in discrete settings.
>
> Existing reMCMC methods rely on decaying step sizes to ensure asymptotic convergence. However, this strategy is practically unsuitable for discrete spaces, particularly in high-dimensional, non-convex discrete landscapes, as discussed in Section 4.2. To address these limitations, we propose a novel swap function (Eq. (10)) specifically designed for discrete sampling. This swap function not only maintains detailed balance but also enables robust exploration across challenging discrete energy landscapes.
>
> Our contributions go beyond a simple integration of existing ideas. By addressing the fundamental incompatibilities of reMCMC with discrete sampling, we advance both the theoretical and practical capabilities of sampling methods, as supported by our theoretical analysis (Section 5) and experimental results (Section 6).
>
> > **What is the dependence of log RMSE on temperature $\tau$ in Ising model results (Fig. 3)? Are the conclusions of Fig. 3 the same for a lower target distribution temperature (e.g., $\tau=0.1$)?**
>
> In our Ising model experiments, the temperatures are typically set to $\tau_1 = 1.0$ and $\tau_2 = 5.0$ for DREXEL and DREAM. At $\tau_1 = 1.0$, the low-temperature sampler draws from an unaltered target distribution in Eq. (1). If $\tau_1 \neq 1.0$, the sampler will produce a biased distribution that deviates from the target distribution, which leads to incorrect sampling results unless a proper rescaling or correction is applied.
>
> When the target temperature is reduced to $\tau = 0.1$, the distribution in Eq. (1) becomes much sharper and more concentrated around the mode. This increased sharpness significantly reduces the sampler's ability to explore the broader energy landscape, making it more prone to getting trapped in local minima. Consequently, it will impair the conclusions drawn in Fig. 3, as the sampler's efficiency in navigating the energy landscape would be diminished.
>
> > **Compared to the Ising model, what if we consider other spin connectivity exhibiting more frustration (e.g. fully connected Sherrington-Kirkpatrick problems)?**
>
> The Sherrington-Kirkpatrick (SK) model is an ideal benchmark to demonstrate the strengths of DREXEL and DREAM due to its fully connected structure and highly frustrated energy landscape. Its rugged landscape with exponentially many local minima requires efficient exploration and avoidance of local traps. DREXEL and DREAM are specifically designed to tackle such complexity, which makes them well-suited for problems like the SK model and aligns with their goal of enhancing exploration in discrete spaces. We appreciate the reviewer’s suggestion and will consider it as a benchmark in our revision.
>
> > **How are temperatures and step sizes chosen in each approach?**
>
> For temperatures, we set $\tau_1 = 1.0$ to sample from the unaltered target distribution. The choice of $\tau_2$ depends on the complexity of the problem. Specifically:
>
> - For simpler landscapes with fewer local minima, $\tau_2$ can be set slightly larger than $\tau_1$ (e.g., $\tau_2 = 2.0$).
> - For more complex landscapes with multiple local minima, $\tau_2$ should be significantly larger than $\tau_1$ (e.g., $\tau_2 = 5.0$ or $10.0$) to enhance exploration and avoid local traps.
>
> For step sizes, the low-temperature step size ($\alpha_1$) in DREXEL and DREAM is set similar to those used in DULA and DMALA to focus on exploring local modes. The high-temperature step size ($\alpha_2$) is larger than $\alpha_1$ to facilitate a broader exploration of the energy landscape. Both step sizes are tuned based on the problem's energy landscape to achieve an effective balance between local exploitation and global exploration.

---

> > ### Author Response · Authors · 2024-11-24
> > **Response to reviewer tAX1 (2/2)**
> >
> > > **Are DMALA and DULA comparisons done at their respective optimal parameters (step size)? How does log RMSE of Ising model results of DMALA and DULA change for step size from 0.1 to 0.5 ?**
> >
> > To the best of our knowledge, DMALA and DULA are conducted at their respective optimal parameter settings.
> >
> > Regarding the log RMSE for the Ising model as step size varies:
> > - DULA achieves optimal performance at step sizes of 0.1 and 0.2, with log RMSE values around -3.0. Beyond this range, performance gradually deteriorates, with log RMSE increasing from approximately -2.1 at a step size of 0.3 to -1.7 at 0.5.
> > - DMALA shows improved performance as the step size increases from 0.1 (log RMSE around -3.1) to 0.3 (around -4.5). Beyond 0.3, the yielded log RMSE values are similar, which remain stable between 0.4 and 0.5 (around -4.5).
> >
> > These observations indicate that DMALA benefits from larger step sizes, while a large step size in DULA introduces extra bias, which deteriorates its performance.
> >
> > > **Could the authors clarify the reasoning behind choosing, e.g., a step size of 0.2 for DULA and 0.4 for DMALA on the Ising model?**
> >
> > The step size of 0.4 for DMALA was chosen to ensure an optimal acceptance rate for the MH step, as recommended in [1]. In contrast, DULA does not include an MH step to enforce detailed balance. To mitigate the bias introduced by its approximate distribution, we reduced the step size. Starting from an initial step size of 0.4, we gradually decreased it and found that the performance reached optimal around 0.2.
> >
> > > **While dDREXEL and bDREAM include a bias, do DREXEL and DREAM incorporate a bias correction as well (Fig. 6)?**
> >
> > We apologize for the confusion. In the experiments presented in the main text, DREXEL and DREAM follow the setup in Eq. (10) and do not include bias correction. However, Fig. 6 represents a specific test where we intentionally introduce an additional bias term into DREXEL and DREAM to evaluate how including such a bias might impact performance. This test was conducted to explore the potential effects of bias correction, but it does not reflect the general setup used in the rest of the experiments.
> >
> > > **How is the bias $\sigma$ in equation (9) chosen in practice?**
> >
> > The bias $\sigma$ in Eq. (9) is selected based on the desired swap rate between the two chains. In practice, we adjust $\sigma$ to achieve a swap rate between 5%-20%, depending on the specific task settings.
> >
> > > **What are the rejection rates for DREAM, and how do step size and temperature choices ($\tau_1$, $\tau_2$) affect it?**
> >
> > In DREAM, we manage rejection rates as follows:
> >
> > - For the low-temperature sampler $( \tau_1$), the rejection rate is controlled around 20%-30%.
> > - For the high-temperature sampler $( \tau_2$), the rejection rate is typically higher (above 50%) and is adjusted based on the complexity of the energy landscape.
> >
> > A higher rejection rate in this chain is expected due to large steps & temperatures to enhance its exploration (the proposed moves are more likely to land in low-probability regions).
> >
> >
> > **References**
> >
> > [1] MCMC Using Hamiltonian Dynamics. arXiv:1206.1901

---

### Official Review · Reviewer_aRLW · 2024-11-04

**Soundness:** 2
**Presentation:** 3
**Contribution:** 2
**Rating:** 3
**Confidence:** 2

**Summary:**

The paper focuses on MCMC samplers for discrete spaces that incorporate both replica exchange and (optionally) a Metropolis correction. Some convergence results (asymptotic and non-asymptotic) are presented, as well as experimental evidence.

**Strengths:**

The empirical results seem promising and suggest that this may indeed be an effective algorithm in practice. In particular, the algorithm clearly outperforms its competitors both when the Metropolis adjustment is and is not present.

The motivation for the approach is clear, and the resulting algorithm is simple and well-principled.

**Weaknesses:**

The theoretical results are not very strong. In particular, Theorem 1 does not provide an non-asymptotic approximation error. Theorem 2 is difficult for me to interpret; see below. In general, these type of replica algorithms are known to be difficult to analyze.

Assumption 3 in the main text is too vague and its presentation in the appendix needs to moved to the main text in order for Theorem 2 to be well-understood.

I have some additional questions regarding Theorem 2 in the main text. See the following section.

**Questions:**

The results seem to be in favour more generally of these type of multiscale chains. Do these methods perform better (empirically) when more levels to this scale are added?

In the figures, do the authors account for the number of rejections? How many times are the proposals typically rejected in practice? Does this affect the optimal step-size scaling compared to the other algorithms?

Can the authors offer more detail on Assumption 4 in the appendix? When does this hold for some reasonable distributions of interest?

Am I correct in noting that Theorem 2 only gives convergence to the biased stationary measure for DREXEL? In this case, can the authors comment on the approximation error of this stationary measure in a non-asymptotic sense?

In Theorem 1, should $\tilde \pi$ and $\pi’$ be referring to the same distribution between the two lines?

147: retains current -> retains the current
In the algorithm, follows -> following

---

> ### Author Response · Authors · 2024-11-24
> **Response to reviewer aRLW (1/2)**
>
> We thank you for your constructive review. Please see our responses below.
>
> > **The theoretical results are not very strong.**
>
> Investigating the quantitative value of the spectral gap for replica exchange Langevin algorithms is indeed a fundamental challenge in the field. Recent work [1] has demonstrated a potential enhancement of the spectral gap in continuous settings when employing replica exchange Langevin diffusion compared to vanilla Langevin diffusion. We anticipate a similar acceleration effect in discrete settings.
>
> Note the current work focuses on the design and implementation of samplers specific to discrete spaces to enhance exploration, we propose this work to bridge practical design with theoretical insights. A deeper exploration of spectral gap quantification in discrete settings is an exciting direction for our future work.
>
> > **Theorem 1 does not provide a non-asymptotic approximation error.**
>
> To clarify, the non-asymptotic approximation error analysis is provided in Theorem 2, which demonstrates the required number of iterations for the empirical distribution to converge to the target distribution under specified error bounds.
>
> The asymptotic analysis in Theorem 1 plays a crucial role in designing the swap function in Eq. (10). This swap function ensures detailed balance and promotes efficient exploration, which is foundational to achieving the mixing improvements analyzed in Theorem 2.
>
> > **Do multiscale chains perform better (empirically) when more levels to this scale are added?**
>
> Empirical evidence from prior works in continuous settings [2, 3] suggests that applying multiple temperature levels in replica exchange Langevin methods can significantly enhance performance, particularly when sampling from multi-modal distributions. This approach is also promising in discrete settings, where multi-scale exploration could address the challenges posed by highly non-convex landscapes.
>
> However, such an effective technique may not directly translate to discrete scenarios. For instance, window-wise corrections in the swap function [2] may require adjustment for discrete sampling, and the efficacy of previously employed swap schemes (stochastic or deterministic even-odd strategies) remains unclear in discrete domains and warrants further investigation.
>
> We recognize that extending multiscale methods to discrete settings involves non-trivial challenges. Exploring the potential benefits and adaptations of these techniques will be a promising direction in our future work.
>
> > **How many times are the proposals typically rejected in practice? Does this affect the optimal step-size scaling compared to the other algorithms?**
>
> The rejection rates are problem-dependent and vary significantly based on the complexity of the energy landscape. For simple tasks, we observed that rejection rates of 30–50% resulted in good performance. This aligns with findings in prior work [4] but does not fully showcase the potential advantages of DREAM.
>
> For non-convex energy landscapes, such as those encountered in training Deep EBMs, the rejection rates differed markedly between the two chains. Specifically, a 20%-30% rejection rate for the low-temperature chain and 70–90% for the high-temperature chain yielded great performance. These settings reflect the need for more exploratory behavior in the high-temperature chain to overcome local traps, while the low-temperature chain focuses on detailed exploitation.

---

> > ### Author Response · Authors · 2024-11-24
> > **Response to reviewer aRLW (2/2)**
> >
> > > **Can the authors offer more detail on Assumption 4? When does it hold for some reasonable distributions of interest?**
> >
> > Assumption 4 states that under a finely discretized space, the expected change in the sampler aligns with the gradient of the potential function, and the expected squared difference reflects the gradient magnitude:
> >
> > $\mathbb{E}[\theta_{i+1} - \theta_i \mid \theta_i] \approx -\frac{\alpha}{2} \nabla U(\theta_i), \quad \mathbb{E}[\|\theta_{i+1} - \theta_i\|^2 \mid \theta_i] \approx \left( \frac{\alpha}{2} \|\nabla U(\theta_i)\| \right)^2 + \beta.$
> >
> > This assumption holds for distributions where the state space is finely discretized and the potential function $U(\theta)$ is smooth. Under these conditions, the discrete sampler's behavior closely approximates continuous Langevin dynamics, which makes the expected change and variance proportional to the gradient and its magnitude. Such scenarios include finely discretized approximations of continuous distributions like Gaussian distributions or Gaussian mixtures.
> >
> >
> >
> > > **Can the authors comment on the approximation error of the stationary measure in a non-asymptotic sense?**
> >
> > The bias in the stationary measure arises from discretization and non-dissipative behavior. Ideally, DREXEL converges to the unbiased stationary measure as the step sizes approach zero ($\alpha_1, \alpha_2 \to 0$) and Assumption 2 becomes strictly dissipative ($\mathscr{B} = 0$).
> >
> > However, in practice, very small step sizes cause DREXEL to become trapped in local minima and lead to slow mixing. To mitigate this, we normally increase step sizes to improve exploration, which introduces discretization bias. This trade-off explains why DREAM (which corrects this bias with additional MH steps) consistently outperforms DREXEL.
> >
> > > **Should $\tilde \pi$ and $\pi'$ be referring to the same distribution between the two lines?**
> >
> > $\tilde{\pi}_n$ refers to the state probabilities of the Markov chain at a specific finite time step $n$ (we omit the subscript here for simplicity), while $\pi'$ represents the stationary distribution, which remains invariant under the transition dynamics of the Markov chain. For an irreducible and aperiodic Markov chain, $\tilde{\pi}$ converges to $\pi'$ as the number of time steps approaches infinity.
> >
> > > **Typoes in Line 14 and Algorithm 1**
> >
> > We appreciate your suggestions on our presentation. We reviewed and corrected these issues in our revised manuscript to ensure clarity and accuracy.
> >
> >
> > **References**
> >
> > [1] Spectral Gap of Replica Exchange Langevin Diffusion on Mixture Distributions. Stochastic Processes and Their Applications 2022.
> >
> > [2] Non-reversible Parallel Tempering for Deep Posterior Approximation. AAAI 2023
> >
> > [3] Non-reversible Parallel Tempering: A Scalable Highly Parallel MCMC Scheme. Journal of the Royal Statistical Society: Series B 2022.
> >
> > [4] MCMC Using Hamiltonian Dynamics. arXiv:1206.1901

---

### Note · Authors · 2024-11-24

**Comment:**

We sincerely thank the anonymous reviewers for their thoughtful and constructive feedback. After careful consideration, we have decided to withdraw our submission to allow sufficient time to address the concerns raised and further improve our work. We deeply appreciate the reviewers’ time and effort in providing valuable insights that will significantly enhance the quality of our research.

**Withdrawal Confirmation:**

I have read and agree with the venue's withdrawal policy on behalf of myself and my co-authors.